

# Comparison of wind-farm control strategies under realistic offshore wind conditions: turbine quantities of interest

Joeri A. Frederik[1], Eric Simley[1], Kenneth A. Brown[2], Gopal R. Yalla[2], Lawrence C. Cheung[2], and Paul A. Fleming[1]

[1]National Wind Technology Center, National Renewable Energy Laboratory, Golden, CO 80401, USA
[2]Wind Energy Technologies Department, Sandia National Laboratories, Albuquerque, NM 87123, USA

**Correspondence:** Joeri A. Frederik (joeri.frederik@nrel.gov)

**Abstract.** Wind farm control is a strategy to increase the efficiency, and therefore lower the levelized cost of energy, a wind farm. This is done by using turbine settings such as the yaw angle, blade pitch angles, or generator torque to manipulate the wake behind the turbine affecting downstream turbines in the farm. Two inherently different wind farm control methods have been identified in literature: wake steering and wake mixing. This paper focuses on comparing the turbine quantities of interest
between these methods for a simple two-turbine wind farm setup, while a companion article (Brown et al., 2025) focuses on the wake quantities of interest for a single wind turbine setup. Both papers use the same set of wind farm simulations based on high-fidelity large-eddy simulations (LES) coupled with OpenFAST turbine models. First, precursor simulations are executed in order to match wind conditions measured with lidars in an offshore wind farm off the US east coast. These measurements indicate general wind conditions that exhibit substantially higher vertical wind shear and veer than any of the
LES studies performed with wind farm control strategies currently available in literature. The precursors are used to evaluate the effectiveness of the control methods. In the LES simulations, the wind veer leads to highly skewed wakes, which has considerable influence on the power uplift of wind farm control strategies. In addition to a baseline controller, four different control strategies, each of which uses either pitch or yaw control, are implemented on the upstream turbine of a simple two-turbine wind farm. Assuming the wind direction is known and constant over time, the simulations show that wake steering is
generally the superior wind farm control strategy considering both wind farm power production and turbine damage equivalent loads (DELs) when substantial wind veer is present. This result is consistent over different wind speeds and wind directions. On the other hand, for similar wind conditions with lower veer, wake mixing was found to yield the highest power production, although at the expense of generally higher loads. This leads us to conclude that the effect of wind veer, which was so far not usually considered, can not be neglected when determining the optimal wind farm control strategy.

## 1 Introduction

Dozens to hundreds of wind turbines are commonly placed closely together in so-called wind farms. While this decreases installation and maintenance costs and increases power capacity for the available installation area, it also comes at a price. As turbines extract energy from the wind, they create a wake in which the wind speed is lower and the turbulence is higher. When





turbines are installed close together in wind farms, some form of interaction between turbines and their wakes is inevitable.
A downstream turbine located partially or fully in the wake of another turbine subsequently produces lower power while experiencing higher loads.

Field studies have shown that the average loss of power on these waked turbines is in the range of 20–40 % with respect to their upstream counterparts (Nygaard, 2014; El-Asha et al., 2017). For low-turbulence wind conditions, this loss can be even higher. As a result, finding the optimal control settings for a wind farm is not necessarily the same as finding the optimal control
settings for individual turbines in the farm. It might be beneficial to have upstream turbines operate at a suboptimal set point for the benefit of downstream turbines – this premise is exploited in the field of wind farm control (WFC) research.

A wind turbine has a limited number of actuators that can be controlled to influence its performance and the characteristics of its wake. Wind turbine control publications focus on one or multiple of the following three actuation methods: blade pitch, generator torque, and nacelle yaw. All of these affect the axial induction factor, i.e., how much of the available kinetic energy
is extracted from the wind, and subsequently, the behavior of the wake. When the blade pitch or the generator torque is moved away from the optimal power coefficient, $C_{P,\max}$, that turbine's power production drops. However, as the turbine extracts less energy from the wind, the kinetic energy in the wake increases, potentially leaving more energy to be extracted by downstream turbines. This method is commonly referred to as *axial induction control*, as it changes the induction factor, i.e., the ratio of the wind velocity reduction in the wake. Similarly, yawing the nacelle at an offset with the wind direction also lowers the power
extraction. Simultaneously, this yaw offset introduces a lateral force on the flow of the wind, thus deflecting the wake away from its usual path. This strategy is known as *wake steering*.

When a fixed yaw offset or derating level is prescribed for given wind conditions, this can be thought of as static or steady-state control strategies. Such strategies aim to find control set points that are constant over time, assuming the wind conditions do not change. Much of the early WFC research focuses on controllers that fall into this categorization. For example, given
a certain wind speed and direction and a predefined wind farm layout, Marden et al. (2012) and Ciri et al. (2017) used optimization algorithms to find the optimal induction factor for each turbine in the farm. While static induction control initially showed some potential, additional studies in higher-fidelity models (Annoni et al., 2016), scaled experiments (Campagnolo et al., 2016), and full-scale experiments (van der Hoek et al., 2019; Bossanyi and Ruisi, 2021) found limited to nonexistent gains. Steady-state wake steering, on the other hand, has shown potential to improve wind farm annual energy production in
full-scale wind farms (Fleming et al., 2019; Howland et al., 2019; Fleming et al., 2020; Doekemeijer et al., 2021) and has since been incorporated in commercial WFC products (Martinez and Coussy, 2024; Bachant et al., 2024).

More recently, interest has increased in time-varying or dynamic WFC strategies. Unlike static strategies, these methods allow or even depend on time-varying control signals under constant wind conditions. A similar distinction between axial induction and wake steering strategies can be made here. Dynamic induction control (DIC), in this paper referred to as the
*pulse* strategy, changes the induction factor of a turbine at a specific frequency to vary the wake width, which has been shown to enhance mixing in the wake of the turbine. This mixing promotes interaction between the slower wind in the wake of the turbine and the faster, freestream flow around it. As a result, the average velocity in the wake increases, and downstream turbines can improve their energy capture. In dynamic wake steering, a similar strategy is applied, but it uses the yaw angle





as the time-varying actuator instead of the induction factor. Research on this strategy is more sparse than on its induction
equivalent (Meyers et al., 2022), and it is found to be less effective than static yaw (Munters and Meyers, 2018b; Howland
et al., 2020). We therefore dismiss dynamic wake steering for now and focus on different variants of DIC.

The easiest way to achieve induction variations is by controlling the pitch actuators. This strategy was first implemented
in wind tunnel experiments executed in Frederik et al. (2020c), showing its effectiveness in a scaled, controlled environment.
Alternatively, a combination of pitch and torque control could be used (Yılmaz and Meyers, 2018). Time variations come in
infinitely many different shapes, but ever since Munters and Meyers (2018a) showed that simple sinusoidal signals are very
effective in DIC applications, most research has focused on sinusoidal variations. Munters and Meyers (2018a) suggested
normalizing the frequency, $f$, used for these sinusoidal variations using the freestream wind speed, $U_\infty$, and the turbine rotor
diameter, $D$, to attain a specific Strouhal number, $St$, defined as

$$St = \frac{fD}{U_\infty},\tag{1}$$

and reported an optimum of $St = 0.25$. Although later publications found slightly different optima (Yılmaz and Meyers, 2018;
Frederik et al., 2020c; Coquelet et al., 2022; Wang et al., 2020), the optimum is always found to be in the range $0.2$–$0.4$.
Differences in optima might be attributed to variations in wind conditions, spacing, actuator implementation, and/or turbine
characteristics. Power uplifts in the range of 2.3–6.3 % are found in these simulation studies for simple two- or three-turbine
wind farms with full wake overlap and low turbulence intensities ($\leq 6\,\%$). However, studies into turbine (damage equivalent)
loads show that the variations in induction, and subsequently rotor thrust, substantially increase the loads experienced by the
excited turbine's tower and blades (Frederik et al., 2020c; Wang et al., 2020; Frederik and van Wingerden, 2022).

To mitigate these large thrust variations while retaining the potential power uplift, a variation to DIC was proposed in
Frederik et al. (2020b). In this publication, the induction was varied over each blade of a turbine individually instead of over
the entire rotor plane. When combined with the Coleman – or multiblade coordinate – transformation (Batchelor and Gill,
1962), dynamic variations in the rotor plane tilt (vertical) and yaw (horizontal) moments can be accomplished. Fleming et al.
(2015) first proposed to apply static offsets in the tilt and yaw moments with the purpose of steering the wake without using a
yaw or tilt angle offset. However, the static application of a multiblade coordinate-transformed offset proved to be ineffective
and was subsequently disregarded.

Nevertheless, when the tilt and yaw moments are varied at the same frequency and with a 90-degree phase offset, a moment
that rotates over the rotor disk over time is created, resulting in a helical wake structure. This method is therefore called
the *helix approach*, where the wake can rotate in either the clockwise or counterclockwise direction, depending on whether
the tilt moment has a 90-degree phase lead or lag with respect to the yaw moment. Frederik et al. (2020b) showed that the
counterclockwise helix enhances wake mixing substantially more than the clockwise helix and found energy uplifts that even
exceeded DIC. Comparing these results to Fleming et al. (2015) once more underlines how dynamic control signals can
outperform static signals, in this case the multiblade coordinate-transformed blade moments, in terms of wind farm power
production.





The helix approach has since attracted more research: Frederik and van Wingerden (2022), Van Vondelen et al. (2023), and Taschner et al. (2023) simulated turbine loads when the helix approach was implemented, finding that the biggest load impact was made on the controlled, upstream turbine. The helix approach was found to impact mostly blade flapwise (or out-of-plane)

moments and tower top moments as well as the pitch bearings due to increased pitching activity. Comparing the helix approach to DIC, Frederik and van Wingerden (2022) found substantially lower tower loads at the expense of the blades and pitch bearings. Taschner et al. (2024) studied the potential benefit of the helix approach compared to wake steering for different wind turbine layouts, demonstrating that the helix could be preferable over wake steering in near-full wake overlap scenarios with turbine spacing of up to 6 rotor diameters. Brown et al. (2022), Korb et al. (2023), and Cheung et al. (2024) investigated

the flow characteristics underlying this method, finding that the helix increases turbulent entrainment, wake deflection, and wake mixing. Coquelet et al. (2024) investigated why the counterclockwise helix is more effective than its clockwise rotating counterpart, and concludes that it is likely due to the wake swirl created by blade rotation. In Van der Hoek et al. (2024) and Mühle et al. (2024), the helix approach was implemented on model turbines in a wind tunnel. The outcomes of these experiments generally agreed with results from simulation studies: The counterclockwise helix outperforms the clockwise

variant, and the obtained power uplift exceeds the gains obtained with DIC in similar wind tunnel experiments (Frederik et al., 2020c; Van der Hoek et al., 2022). Finally, Van Den Berg et al. (2022) investigated the implementation of the helix approach on a floating offshore wind turbine and found that the combination of pitch actuation and platform motion might further enhance the wake mixing effect.

The same multiblade coordinate-transformed tilt and yaw moments can also be applied separately (Frederik et al., 2020a;

Cheung et al., 2024), or at higher harmonics (Huang et al., 2023). These approaches have not yet been investigated to the same degree as the helix approach, but initial results are promising and show that perhaps strategies based on individual pitch control (IPC) other than the helix approach should be considered in future research.

Because the dynamic pitch methods work on the premise that they actively increase the mixing between the slower flow in the wake and the faster flow around the wake, we will reference the totality of these strategies as active wake mixing (AWM)

strategies. In this paper, the validity of these relatively novel AWM strategies is further investigated by comparing them, in terms of both power uplift and structural loads, to the more established wake steering strategy. We implement the helix approach, an IPC approach where only the yaw moment is varied, DIC (or, the pulse strategy), and wake steering on a wind turbine, and examine the effect they have the turbine, on the wake, and on a potential downstream turbine. For this purpose, high-fidelity flow simulations are set up to resemble data acquired from lidar measurements in a real offshore wind farm off the east coast

of the United States. We investigate the effect that wind properties such as speed, direction, turbulence intensity, and veer have on the effectiveness of WFC strategies. Apart from wind turbine and wind farm power production, we also review the damage equivalent load (DEL) of multiple critical turbine components. The results of these simulations contribute to our understanding of the potential of AWM as a WFC strategy depending on wind speed, turbulence intensity, atmospheric stability, and wind farm layout.

This paper is a companion to Brown et al. (2025). Where this paper mainly focuses on the turbine-level behavior of the different control strategies, Brown et al. (2025) dives deeper into the fluid dynamics behind the results. On multiple occasions,





we will summarize results from and reference Brown et al. (2025) for the reader's convenience. However, for a complete understanding of the effect AWM has on both turbine and flow, the reader is advised to study both papers.

## 2 Methodology

As mentioned in Sect. 1, the objective of this paper is to compare different AWM strategies to the more established wake steering control approach. To assess the effectiveness in terms of wind farm power and turbine structural loads of these strategies, high-fidelity simulations of the turbine-flow interaction are executed. In this section, we discuss the simulation environment in detail as well as the control strategies that are implemented. First, we review the simplified wind farm setup used in this study.

### 2.1 Wind farm setup

As discussed previously, applying WFC only makes sense if there is some sort of interaction between turbines through their wakes. Without interaction, the optimal WFC strategy in terms of power production would always be to have each turbine operate at its individual optimum. However, wind farm layouts are not standardized but depend on the topology of the farm, and wind directions vary over time. As a result, the potential of WFC varies substantially between different wind farms, and it is not possible to assess one specific case and then extrapolate the results for different farms. We therefore choose to study a simplified wind farm case with only two interacting turbines. The internal distance between the two turbines is set to be $5D$ in the streamwise direction, assuming perfect alignment with the average wind direction, unless stated otherwise.

Apart from these two-turbine simulations, we also execute simulations with a single turbine to study the wake of the turbine. By sampling flow data in the wake, we are able to estimate the potential power uplift of a downstream turbine at any given location. We call this a virtual turbine and determine its power by using the rotor-averaged wind speed, $\bar{U}_r(t)$, as a function of wind speed data points $u_{i,j,k}(t)$, where $i,j,k$ are the grid points in the $x$-, $y$- and $z$-directions, respectively, and virtual hub location $\boldsymbol{x} = [x_h, y_h, z_h]$.

The set of points within the rotor-swept area, $C$, is first defined as

$$C(y_h, z_h) = \left\{ (j,k) \mid (y_j - y_h)^2 + (z_k - z_h)^2 \leq (D/2)^2 \right\}, \tag{2}$$

where $y_h$ and $z_h$ are the location of the hub of this virtual turbine in the lateral and wall-normal (vertical) directions, respectively. Note that the location $x_h$ of the turbine in the streamwise direction does not affect $C$, as we assume the virtual turbine is yawed perpendicular to the average wind direction. The rotor-averaged wind speed at streamwise location $x_h$ is then defined as

$$\bar{U}_r(t, \boldsymbol{x}) = \frac{\sum_n \sum_m u_{h,j,k}(t)}{|C(t_h, z_h)|} \qquad \forall \quad (m,n) \in C(y_h, z_h), \tag{3}$$

where $|C(y_h, z_h)|$ is the size of set $C(y_h, z_h)$. Finally, the power of the virtual turbine, $P_v(t, \boldsymbol{x})$, is computed as

$$P_v(t, \boldsymbol{x}) = \frac{1}{2} \rho A_r C_P \left( \bar{U}_r(t, \boldsymbol{x}) \right) \left( \bar{U}_r(t, \boldsymbol{x}) \right)^3, \tag{4}$$





where the power coefficient, $C_P$, is assumed to be a function of the wind speed as provided by a lookup table of the chosen turbine model (see Sect. 2.3.2). In Sect. 3, we evaluate how well the power of a virtual turbine corresponds to simulations with an actual downstream turbine model.

## 2.2 Control strategies

In this paper, we investigate five different WFC strategies. Before elaborating on these strategies, it is relevant to discuss how a standard wind turbine controller works. In below-rated wind conditions a turbine controller typically aims to maximize $C_P$, so it operates using constant blade pitch angles and a nacelle yaw angle that is aligned with the wind direction. The generator torque is then controlled to regulate the blade tip-speed ratio, $\lambda$, toward its optimum for maximal power extraction, $C_P^{\mathrm{opt}}(\lambda)$. This strategy can be called greedy control from the perspective of individual wind turbines.

In above-rated conditions, maximal power production is no longer the control objective. In this regime, a typical wind turbine controller actuates the blade pitch angles to keep the rotor speed constant while producing rated power. As a result, the controller is pitching the blades to counteract changes in wind speed.

In this paper, the control strategy aimed at maximizing the power yield of individual turbines is the first approach implemented. It is used as the baseline strategy to which we compare the different WFC strategies. Furthermore, all downstream
turbines implemented in this paper are also controlled using this baseline controller.

The four WFC strategies investigated in this paper are summarized in Table 1. Three of these strategies are AWM strategies: DIC, from here on out referred to as the pulse strategy, which uses collective pitching, and the helix and side-to-side strategy, using IPC. These AWM strategies are compared to wake steering, which uses a time-invariant yaw angle offset. Details on how these WFC strategies are implemented can be found in Sect. 2.3.3.

**Table 1.** Overview of WFC strategies and how they are applied on the upstream wind turbine.

| Control strategy | Actuation | Desired effect |
|---|---|---|
| Baseline | None | Functioning as comparison to the different WFC strategies |
| Helix | Individual blade pitch variations | Dynamically steering the wake horizontally and vertically |
| Pulse | Collective blade pitch variations | Dynamically varying the rotor thrust to modulate wake depth |
| Side-to-side | Individual blade pitch variations | Dynamically steering the wake horizontally |
| Wake steering | Nacelle yaw offset | Statically steering the wake horizontally |

## 2.3 Simulation environment

The simulations conducted for this paper have been executed in AMR-Wind (Lawrence Berkeley National Laboratory et al., 2024), an incompressible flow solver for wind turbine and wind farm simulations that is part of the ExaWind ecosystem. AMR-Wind is actively being developed by Lawrence Berkeley National Laboratory, National Renewable Energy Laboratory





(NREL), and Sandia National Laboratories, and executes large-eddy simulations (LES) of atmospheric boundary layer flows.
For more information on AMR-Wind, see, e.g., Brazell et al. (2021) and Min et al. (2024).

### 2.3.1 Flow conditions

The wind flow conditions of the simulations used in this paper are based on measurements from a floating-lidar campaign at a proposed wind farm site in the New York Bight (Mason, 2022). Wind measurements were collected over a period of 1.6 years, with the lidars sampling at heights between 20 and 200 m. All data used to set up these simulations are publicly available
(DNV, 2023).

The effectiveness of different WFC strategies can depend heavily on the wind conditions. Therefore, the lidar data are divided over three wind speed (WS) and turbulence intensity (TI) bins: low ($6\,\mathrm{m\,s^{-1}} < \mathrm{WS} \leq 7\,\mathrm{m\,s^{-1}}$), medium ($8.5\,\mathrm{m\,s^{-1}} < \mathrm{WS} \leq 9.5\,\mathrm{m\,s^{-1}}$) and high ($11\,\mathrm{m\,s^{-1}} < \mathrm{WS} \leq 12\,\mathrm{m\,s^{-1}}$) WS cases, and low (TI $\leq 5\,\%$), medium ($5\,\% < \mathrm{TI} \leq 10\,\%$), and high (TI $> 10\,\%$) TI cases, see Brown et al. (2025) details. This results in a total of nine cases that could be investigated. Both
the low- and medium-WS cases are in the below-rated regime of the modeled turbine, while the high-WS cases are in the above-rated regime (see Sect. 2.3.2).

Note that WFC is generally considered to be most effective at low TI. Turbulence provides mixing between the slower flow in the wake with the faster freestream flow, thus inducing natural wake recovery. As WFC reduces wake losses, lower natural losses imply lower potential for improvement by implementing WFC. Furthermore, above-rated conditions are less
favorable for the implementation of WFC, as upstream turbines are already operating at lower-than-maximal power and thrust coefficients. Subsequently, the relative wake deficit – and therefore the potential gain for WFC – is lower than in below-rated conditions. As a consequence, we focus most of our attention in this paper on the low- and medium-WS, low-TI cases. Nevertheless, we also study the high-WS, low-TI, and medium-WS, medium-TI cases, to demonstrate their effects on WFC strategies. More details on the setup of these simulations can be found in the companion paper Brown et al. (2025).

In addition to the simulation cases based on lidar measurements, one more simulation case is added to study the direct effect of veer and shear on the effectiveness of WFC. This case is set up to be comparable to the main medium-WS, low-TI case, but with negligible veer and shear. There is some evidence in literature (Bodini et al., 2017; Churchfield and Sirnivas, 2018) that LES overestimates the amount of wake skewing induced by wind veer, which is discussed in more detail in Sect. 3.3.2. This case is therefore included to account for that possibility. Furthermore, it serves to show the direct effect of both TI (by
comparing to the medium-TI case) and wake skewing (by comparing to the low-TI case) on the effectiveness of WFC strategies. The numerical domain of simulation for all cases is summarized in Table 2, and Table 3 shows relevant quantities that define the air flow through the domain in all of the simulation cases.

### 2.3.2 Turbine model

The AMR-Wind flow solver described above is coupled with the well-known OpenFAST wind turbine simulation tool (Na-
tional Renewable Energy Laboratory, 2024b). OpenFAST couples computational models for aerodynamics, electrical system dynamics, structural dynamics, and controls to simulate the wind turbine response in the time domain.



**Table 2.** An overview of the most relevant AMR-Wind LES settings used in the different wind condition cases reviewed in this paper.

| Case | Simulation length | Domain size | Cell size (outer region) | Cell size (near rotor) | Air density |
|---|---|---|---|---|---|
| Low WS, low TI | 1000 s | $6.72 \times 2.0 \times 0.96$ km | $5 \times 5 \times 5$ m | $2.5 \times 2.5 \times 2.5$ m | $1.2456$ kg m$^{-3}$ |
| Medium WS, low TI | 1200 s | $4.56 \times 2.0 \times 0.96$ km | $5 \times 5 \times 5$ m | $2.5 \times 2.5 \times 2.5$ m | $1.2456$ kg m$^{-3}$ |
| Medium WS, low TI, low veer | 1200 s | $5.12 \times 5.12 \times 1.28$ km | $10 \times 10 \times 10$ m | $2.5 \times 2.5 \times 2.5$ m | $1.225$ kg m$^{-3}$ |
| Medium WS, medium TI | 1000 s | $7.2 \times 4.0 \times 1.44$ km | $10 \times 10 \times 10$ m | $2.5 \times 2.5 \times 2.5$ m | $1.2456$ kg m$^{-3}$ |
| High WS, low TI | 1000 s | $6.72 \times 2.0 \times 0.96$ km | $5 \times 5 \times 5$ m | $2.5 \times 2.5 \times 2.5$ m | $1.2456$ kg m$^{-3}$ |

**Table 3.** Wind conditions for different precursors, including WS, TI, and veer levels. The veer is expressed in terms of degrees of wind direction change over the rotor disk (height of 30 m to 270 m). The shear is expressed as the power exponent that best fits the vertical wind speed profile and, in brackets, as the wind speed difference over the rotor disk in meters per second.

| Case name | Acronym | Hub height wind speed | Hub height TI | Veer | Shear |
|---|---|---|---|---|---|
| Low WS, low TI | LSLT | $6.48$ m s$^{-1}$ | $2.85\%$ | $8.77°$ | $0.139$ ($1.75$ m s$^{-1}$) |
| Medium WS, low TI | MSLT | $9.00$ m s$^{-1}$ | $2.36\%$ | $11.86°$ | $0.177$ ($3.03$ m s$^{-1}$) |
| Medium WS, low TI, low veer | MSLT-LV | $8.97$ m s$^{-1}$ | $3.36\%$ | $1.99°$ | $0.080$ ($1.56$ m s$^{-1}$) |
| Medium WS, medium TI | MSMT | $8.98$ m s$^{-1}$ | $7.00\%$ | $1.32°$ | $0.070$ ($1.30$ m s$^{-1}$) |
| High WS, low TI | HSLT | $11.47$ m s$^{-1}$ | $2.84\%$ | $7.18°$ | $0.163$ ($3.60$ m s$^{-1}$) |

Because this study focuses specifically on offshore wind conditions, the 15 MW monopiled offshore reference wind turbine model developed in IEA Wind Task 37 is used (Gaertner et al., 2020). This turbine is henceforth referenced to as the IEA 15 MW turbine, and its properties are defined in Table 4.

**Table 4.** Properties of the IEA 15 MW turbine.

| Parameter | Value |
|---|---|
| Rated power | 15 MW |
| Turbine class | IEC Class 1B |
| Number of blades | 3 |
| Cut-in wind speed | $3$ m s$^{-1}$ |
| Rated wind speed | $10.59$ m s$^{-1}$ |
| Cut-out wind speed | $25$ m s$^{-1}$ |
| Minimum rotor speed | $5.0$ rpm |
| Maximum (rated) rotor speed | $7.56$ rpm |
| Hub height | $150$ m |
| Rotor diameter | $240$ m |





### 2.3.3 Turbine controller

To implement the different control strategies on the IEA 15 MW turbine, NREL's reference open-source controller (ROSCO v2.8.0; National Renewable Energy Laboratory (2024a)) is used. ROSCO was developed to offer the scientific community a baseline wind turbine reference controller with industry-standard functionality (Abbas et al., 2022).

For the work executed in this paper, AWM functionality had to be added to ROSCO. This functionality was included in ROSCO version 2.8.0, and uses the Coleman transformation (Batchelor and Gill, 1962) to set the parameters for different wake mixing strategies. The pitch angles $\theta_i$, $i \in 1, 2, 3$ are defined as the outcome of the inverse Coleman transformation:

$$\begin{bmatrix} \theta_1(t) \\ \theta_2(t) \\ \theta_3(t) \end{bmatrix} = \begin{bmatrix} \cos(n\psi_1(t)) & \sin(n\psi_1(t)) \\ \cos(n\psi_2(t)) & \sin(n\psi_2(t)) \\ \cos(n\psi_3(t)) & \sin(n\psi_3(t)) \end{bmatrix} \begin{bmatrix} M_{\text{tilt}}(t) \\ M_{\text{yaw}}(t) \end{bmatrix}, \tag{5}$$

where $n$ is the Coleman transformation harmonic number, and $\psi_i(t)$, $i \in 1, 2, 3$ is the azimuth position of blade $i$ at time $t$. In ROSCO, the user is able to define $M_{\text{tilt}}$ and $M_{\text{yaw}}$ individually as a constant or sinusoidal signal:

$$M = A\cos(2\pi f_e t + \phi) = A\cos(\omega_e t + \phi), \tag{6}$$

with amplitude $A$, frequency $f_e$, and offset angle $\phi$ user inputs, and $\omega_e = 2\pi f_e$. For the AWM strategies investigated in this paper, the required settings are summarized in Table 5. Note that, equivalent to the Coleman transform method, the AWM strategies can also be defined by using the normal mode method (Cheung et al., 2024). For completeness, these settings are shown in Table 5 as well.

To limit the number of simulations to be executed, the amplitude and frequency of the AWM strategies is not optimized for all wind conditions. Instead, a Strouhal frequency of $St = 0.3$ and a pitch amplitude of $A = 4$ is used in all simulations. This frequency was found to be optimal in the MSLT case, and from literature (?Van der Hoek et al., 2024) we know that the gradient is low around the optimum. The pitch amplitude of $A = 4$ is relatively large, but is chosen to represent the case in which maximizing power production is prioritized over minimizing loads. Decreasing the amplitude has a close-to-linear effect on power production and DEL, see, e.g., Van Vondelen et al. (2023). Equivalently, a relatively large yaw angle offset of $20°$ is used in all simulations.

As the azimuth angle is a function of rotor speed $\omega_r$,

$$\psi_i(t) = \omega_r t + \frac{2\pi}{3}(i-1), \tag{7}$$

the individual pitch angles $\theta_i$ for each blade $i$ are a product of two sinusoidal signals, which, using simple rules of trigonometry, can be simplified to:

$$\begin{aligned} \theta_i^{\text{pulse}} &= A\sin(\omega_e t), \\ \theta_i^{\text{helix}} &= A\sin\left((\omega_r + \omega_e)t + \frac{2\pi}{3}(i-1)\right), \\ \theta_i^{\text{side-to-side}} &= A\sin\left(\omega_r t + \frac{2\pi}{3}(i-1)\right)\sin(\omega_e t). \end{aligned} \tag{8}$$





**Table 5.** Control settings for AWM as implemented in ROSCO. $St$ is the Strouhal number, set at 0.3 in this paper, and $D = 240\,\mathrm{m}$ is the turbine rotor diameter.

| Control strategy | General settings | | | Coleman method | | | Normal mode method | |
|---|---|---|---|---|---|---|---|---|
| | $\mathbf{A}_{\text{tilt}}$ | $\mathbf{A}_{\text{yaw}}$ | $\mathbf{f_e}$ | $\mathbf{n}$ | $\phi_{\text{tilt}}$ | $\phi_{\text{yaw}}$ | $\mathbf{n}$ | $\phi_{\text{clock}}$ |
| Pulse | 4 | n/a | $St \cdot U_\infty / D$ | 0 | $0°$ | n/a | 0 | $90°$ |
| Helix | 4 | 4 | $St \cdot U_\infty / D$ | 1 | $0°$ | $90°$ | $-1$ | $90°$ |
| Side-to-side | 0 | 4 | $St \cdot U_\infty / D$ | 1 | n/a | $0°$ | $+1, -1$ | $90°$ |

Note that the rated rotor rotational frequency $f_r = 7.56\,\mathrm{rpm} = 0.126\,\mathrm{Hz}$ is on the order of magnitude of 10 times faster than the standard excitation frequency $f_e = St \cdot U_\infty / D = 1.25 \cdot 10^{-3} \cdot U_\infty$. As a result, the pitch frequency of the pulse strategy is about 10 times slower than that of the helix and side-to-side strategies. This is visualized in Fig. 1, which shows the pitch signals of
one blade for all control strategies.

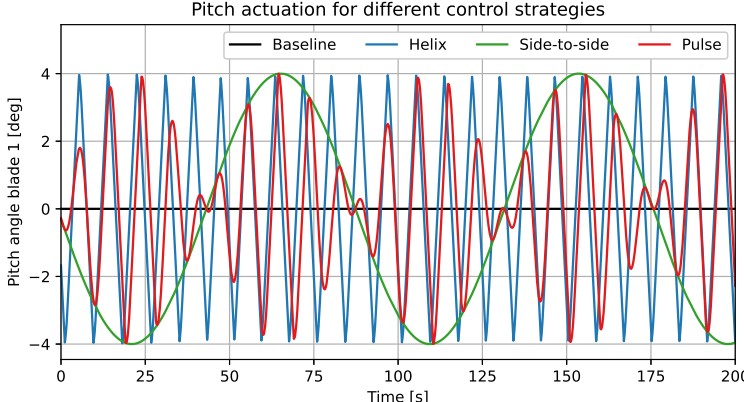

**Figure 1.** A visualization of typical pitch actuation for a single blade when different AWM strategies are applied, as defined in Eq. (8). This example comes from a simulation with a wind speed $U_\infty = 9.0\,\mathrm{m\,s^{-1}}$, with an amplitude of $A = 4°$ and a Strouhal number of $St = 0.3$, resulting in an excitation frequency $f_e = 1.13 \cdot 10^{-2}\,\mathrm{Hz}$ and an excitation period of $T_e = 88.9\,\mathrm{s}$. The average rotor speed over these simulations is $f_r = 6.6\,\mathrm{rpm} = 0.11\,\mathrm{Hz}$.

Finally, for the wake steering case, no active control is implemented. For this strategy, the turbine runs using the same control settings as the baseline case, with the exception of a constant yaw misalignment defined with respect to the mean wind direction. The positive yawing direction is defined as counterclockwise when seen from above.

## 2.4 Load calculations

A common argument that is regularly made against WFC strategies, specifically AWM, is that the benefits in terms of power increase do not outweigh the drawback of higher structural turbine fatigue loads. To address these valid concerns, we will





not only study wind farm power uplift in this paper but also investigate the effect on fatigue loads. Specifically, we look at a quantity called damage equivalent load (DEL).

The DEL of a component is calculated using rainflow counting to obtain the amplitudes and frequencies of different load
cycles within load bins $k$. To correct for mean loads, the Goodman correction (see, e.g., Sutherland and Mandell (2004)) is applied to obtain stress signals $L$ based on the ultimate stresses of the materials. The DEL is then computed as

$$\text{DEL} = \left( \frac{1}{T} \sum_k n_k \left( L_k \right)^m \right)^{\frac{1}{m}},  \tag{9}$$

where $T$ is the number of 1 Hz cycles over the simulation, $n_k$ is the number of cycles of amplitude $L$ in load bin $k$, and $m$ is the slope of the Wöhler curve of the component in question. All material properties used in these calculations are shown in
Table 6. Note that when a component consists of multiple different materials, the lower ultimate load is used here to obtain a worst-case DEL.

**Table 6.** Material properties of all the components considered in the load studies executed in this publication.

| Component | Ultimate stress | Wöhler exponent |
|---|---|---|
| Blade | 1047 MPa | 10 |
| Tower | 450 MPa | 4 |
| Yaw bearing | 113 MPa | 4 |
| Low-speed shaft | 814 MPa | 4 |

In turbine load calculations, orthogonal signals – such as the blade root flapwise and edgewise bending moments – are typically studied individually. However, in reality, these orthogonal projections form a single moment that has both a magnitude and a direction. Therefore, in Thedin et al. (2024), a method is used that combines these orthogonal loads over a 180° rose with
10° steps. The DEL is then calculated for each of the 18 directional bins, and the highest DEL is taken as the critical DEL. The loads of the components described in Table 6 are determined by using the blade root flapwise and edgewise moments, the tower base fore-aft and side-side moments, the yaw bearing pitch and roll moments, and the low-speed shaft nonrotating yaw and tilt bending moments at the shaft tip, respectively.

## 3 Results

In this section, we discuss the results from the simulations executed based on the methodology described in Sect. 2. Unless stated otherwise, the simulations consist of a two-turbine wind farm, with $5D$ streamwise spacing and $0D$ lateral spacing, i.e., the downstream turbine is experiencing full wake overlap in baseline operation.





## 3.1 Upstream turbine behavior

First, we consider the behavior of the upstream turbine, on which the different control strategies introduced in Sect. 2.2 are
implemented. Some relevant turbine signals are plotted in Fig. 2. As known from literature, the most significant power drop is
observed for the wake steering case. However, it is also expected to yield the most significant uplift on downstream turbines.
When the pulse is implemented, the variations in thrust also lead to power fluctuations, as would be expected. Note that these
power fluctuations are not present for the IPC-based AWM strategies, as these are designed to keep the rotor-averaged induction
constant.

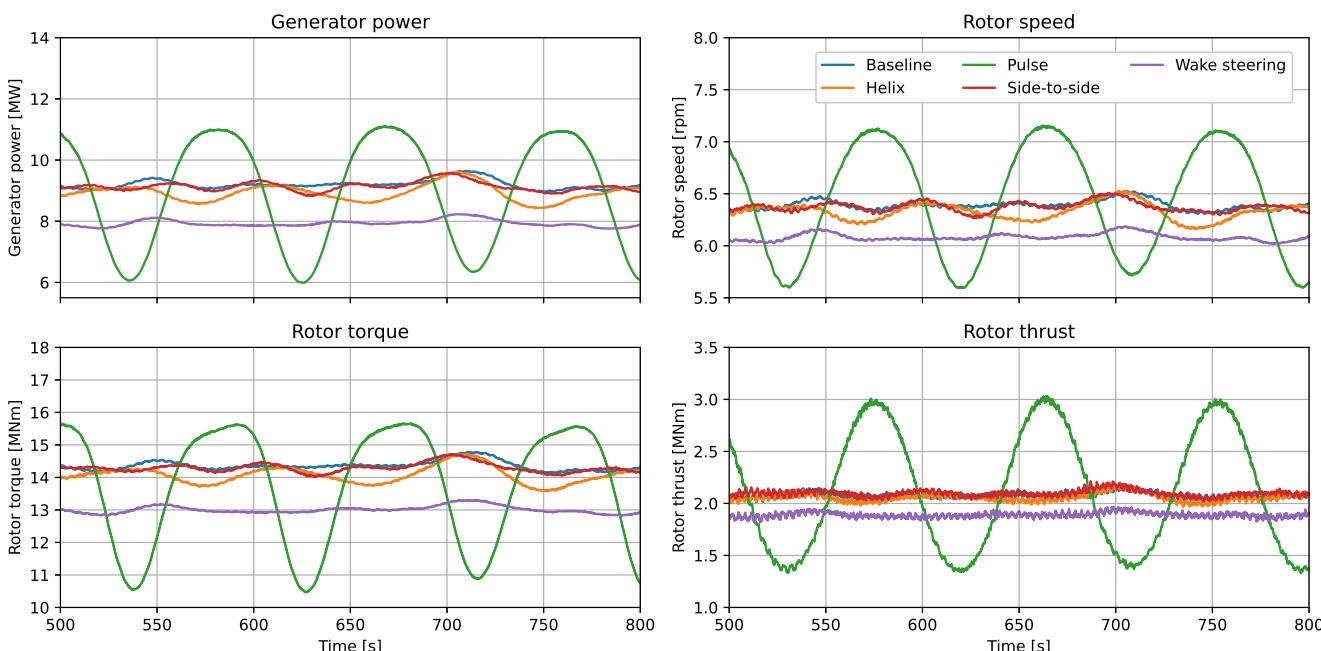

**Figure 2.** The generator power and rotor speed, torque, and thrust for all five control cases described in Table 1. The active wake control
strategies each have an amplitude of $4°$, while the wake steering yaw offset is $+20°$.

The average power production of the turbine for all control strategies is shown in Table 7. The power loss of the AWM
strategies is well within single digit percentages; the helix strategy has the highest power loss, and the side-to-side strategy has
the lowest power loss.

## 3.2 Single-turbine wake analysis

Next, we study the wake behind a single turbine. In Sect. 3.2.1, we investigate the effect the wind veer has on wake skewing
in baseline control operation, and in Sect. 3.2.2, the effect of the different control strategies on the wind speed in the wake is
examined.



**Table 7.** The average power production over the last eight excitation periods ($\approx 707\,\mathrm{s}$) of the single turbine LES simulation, for all five control strategies. The relative loss is given compared to the baseline case.

| Control strategy | Average power | Relative loss |
|---|---|---|
| Baseline | 9.12 MW | n/a |
| Helix | 8.83 MW | $-3.22\,\%$ |
| Pulse | 8.91 MW | $-2.27\,\%$ |
| Side-to-side | 9.05 MW | $-0.76\,\%$ |
| Wake steering | 7.87 MW | $-15.9\,\%$ |

### 3.2.1 Wind veer analysis

As shown in Table 3, the low-TI cases all exhibit substantial wind veer. Specifically, case MSLT has very high veer: $11.86°$ over the $D = 240\,\mathrm{m}$ rotor disk. As a result, we expect the wake to be skewed, possibly affecting the downstream wake loss. To estimate the level of wake skewing, the average wake profile is fitted to a Gaussian curve defined as

$$f(x) = U_\infty - U_d \cdot e^{-\frac{(r-\mu)^2}{2\sigma^2}}, \tag{10}$$

where $U_d$ is a measure of the wake deficit, $r$ is the lateral distance from the hub center, $\mu$ is the lateral wake center position, and variance $\sigma^2$ is a measure for the wake width. Using simple nonlinear least squares optimization to find all the above-mentioned parameters, we can extract the wake center $\mu$ at different elevations. The result is shown in Fig. 3 for the MSLT case, at $5D$ behind a turbine operating with the baseline controller.

Considering Fig. 3, the wake skew at $5D$ is substantial, but in line with what could be expected based on the veer (solid blue line). Apart from the exact skewing of the wake, other factors might affect the power production of a downstream turbine. First, the shape of the wake is not circular or even very elliptical. Instead, the wake width stays relatively consistent over the $y$-axis while only the wake center shifts. As a result, the wake would not affect a downstream turbine as much as would be the case in unveered conditions, and subsequently, the potential gain of using WFC is also expected to be lower. Second, the aforementioned wake center does not align with the wind direction but instead veers slightly to the right looking downstream. At hub height, this offset is $0.12D$ ($28.7\,\mathrm{m}$), equivalent to a wind direction offset of $1.4°$. Both effects could possibly favor wake steering over AWM as a WFC strategy, as both make it easier to steer the deepest part of the wake away from downstream turbines.

Figure 4 similarly shows the wake skewing at $5D$ downstream for the low- and high-WS simulations. In these cases, the veer, and subsequently the wake skew, is slightly less prominent than in the MSLT case. The correspondence between the veer and the wake skew is still close to a 1:1 ratio. To see if this is realistic, we compare the wakes to similar cases found in literature. In Churchfield and Sirnivas (2018), the effects of wake skew were studied using a different LES code called SOWFA (Churchfield and Lee, 2012). This study investigated the actual-to-expected skew angle ratio, i.e., the ratio between skew and veer, as a function of downstream location, and shows a ratio close to 1 at $2D$ downstream, going down to 0.8 at $10D$. This is

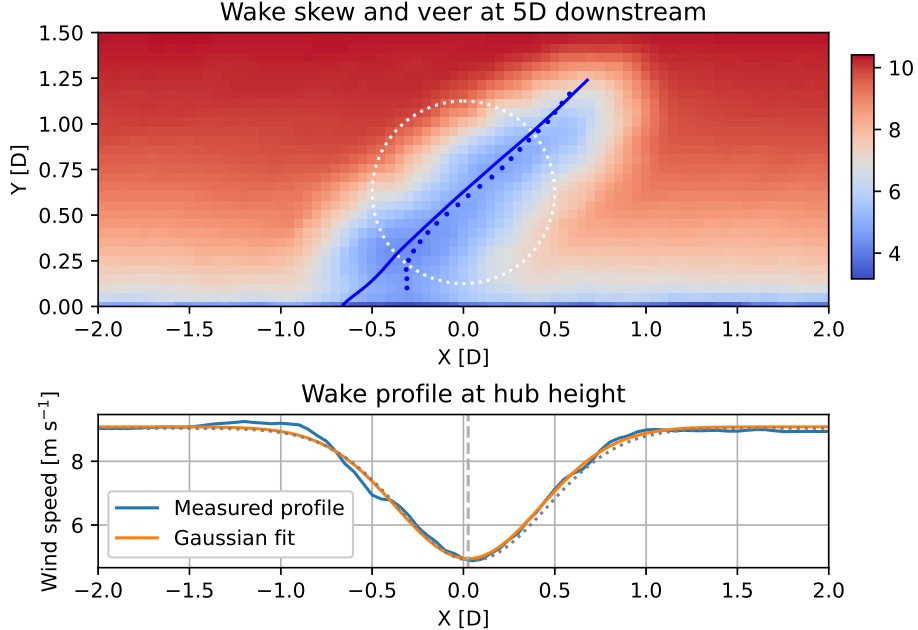

**Figure 3.** Average wake skewing at $5D$ downstream of a turbine operating using baseline control on the MSLT simulation. The average is taken over the last 500 s of the simulation. In the top figure, the velocity profile over a cross section of the flow is shown. The solid line shows the expected skew, assuming veer is 1:1 correlated to skew. The dotted line shows the estimated wake center based on the Gaussian fits. The white dotted circle represents the location of the upstream rotor disk. The bottom figure shows the average wind speeds and the corresponding fit and resulting wake center at hub height ($y/D = 0.625$).

very similar to the LSLT case, as shown in Fig. 5a. The MSLT and HSLT cases have slightly higher relative skewing, which is likely to influence the performance of downstream turbines. Churchfield and Sirnivas (2018) showed that a turbine located $9D$ downstream of a different turbine in a skewed wake produces 30 % more power than in a nonskewed wake, and experiences generally higher DELs.

Bodini et al. (2017) studied the structure of wakes using lidar wind measurements in a land-based wind farm. This study showed that, at the measured downstream distance of $8.5D$ (680 m), the wake skewing was substantially lower than a 1:1 relationship with wind veer (see the solid lines in Fig. 5b). The values at $8.5D$ downstream for the simulation cases from this paper are shown by the dots. In comparison, the simulation cases presented here, as well as those in Churchfield and Sirnivas (2018), have wake skewing that is much closer to the wind veer conditions. Furthermore, in the field measurements,

the relationship effect of veer on wake skewing decreases as veer increases, whereas this is not observed in any of the LES cases.

Churchfield and Sirnivas (2018) showed the effect that wake skewing has on downstream turbine behavior, and Bodini et al. (2017) showed that the wake skew in our simulation cases is substantially higher than that observed in field experiments.





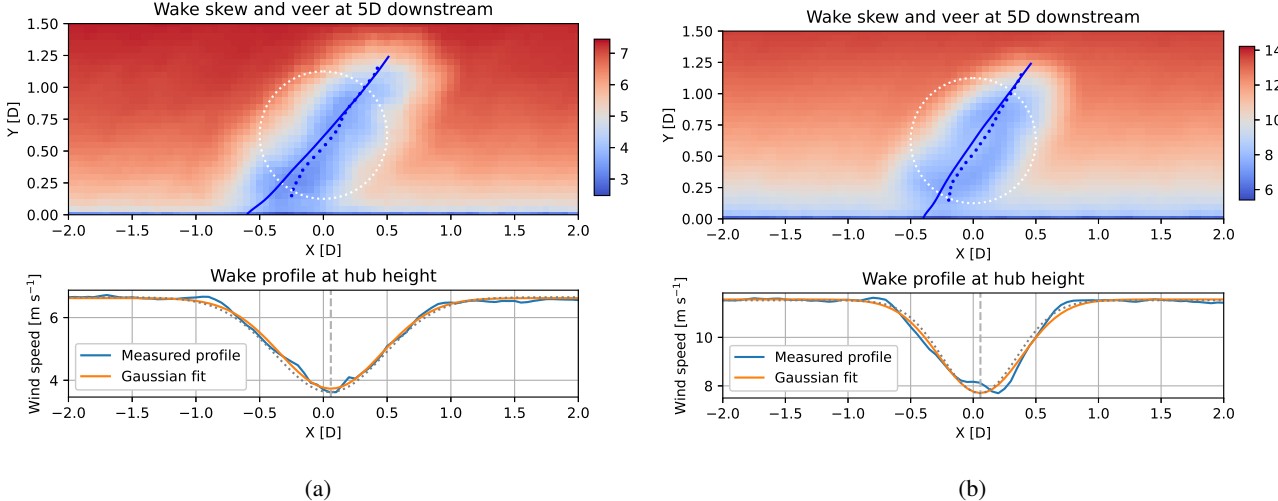

**Figure 4.** Average wake skewing at $5D$ downstream of a turbine operating using baseline control on the LSLT (a) and HSLT (b) simulations. The average is taken over the last $500\,\mathrm{s}$ of the simulation. In the top figure, the velocity profile over a cross section of the flow is shown. The solid line shows the expected skew, assuming veer is 1:1 correlated to skew. The dotted line shows the estimated wake center based on the Gaussian fits. The white dotted circle represents the location of the upstream rotor disk. The bottom figure shows the average wind speeds and the corresponding fit and resulting wake center at hub height ($y/D = 0.625$).

Therefore, the results in this paper might overestimate the effect of wind veer on WFC. However, these field experiments were

conducted onshore, whereas the wind measurements used to generate these simulations were taken offshore. It is therefore also plausible that the wake skewing effect is bigger offshore than it is onshore due to a lack of ground effects to disturb the flow.

### 3.2.2 Energy in the wake

In this section, we study the level of wake recovery realized by the different control strategies, and the effect it could have on wind farm power production. Figure 6 shows the rotor-averaged wind speed for a downstream turbine location, assuming

alignment with the wind direction, as a function of downstream distance. Note that this figure shows that the streamwise plane at hub height does not give an accurate estimate of the rotor-averaged wind speed for all cases. There is a clear discrepancy with the more accurate cross sections that is likely explained by the significant wake skew present in these cases. Furthermore, this figure clearly shows that wake steering results in the highest velocity uplift in the wake. However, this strategy also accounts for the largest upstream power loss.

To get the full picture in terms of wind farm power, the power loss at the upstream turbine should also be taken into account. This is shown in Fig. 7, where the *combined* power production of a two-turbine wind farm is estimated for any position of the downstream turbine, based on the rotor-averaged velocity obtained from the cross-sectional planes. This figure shows that for specific locations of the downstream turbine, wake steering and the pulse have the highest predicted power uplift, 7.7 % and 9.0 %, respectively. Both these optima are at $4D$ downstream and have a slight offset – $0.4D$ to the right for the pulse, $0.3D$

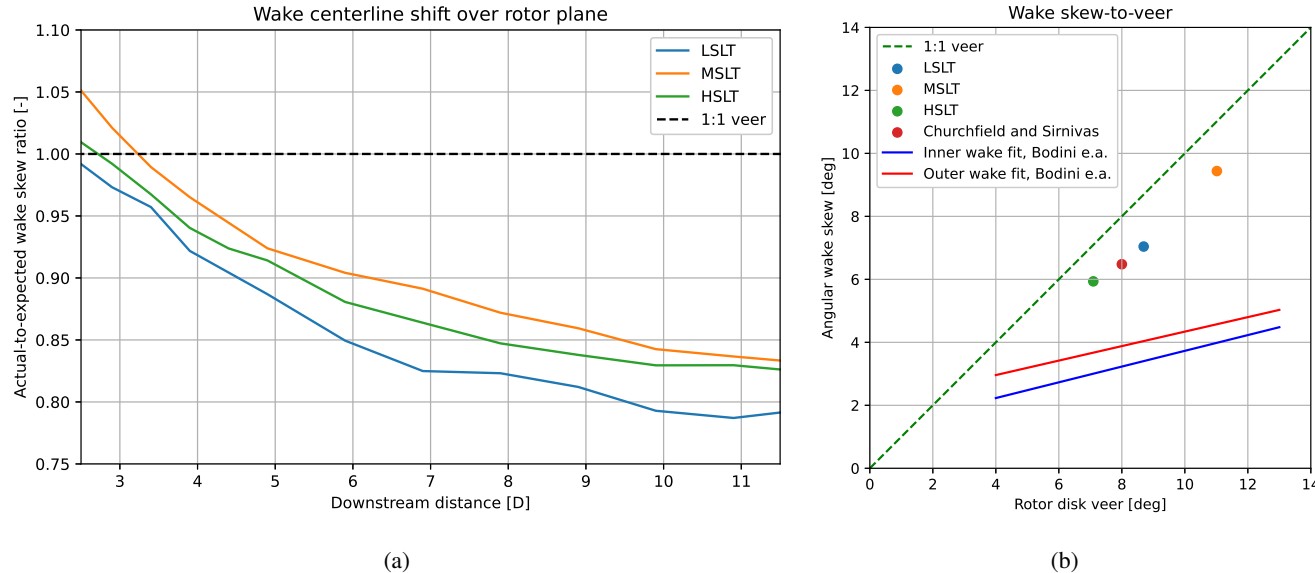

(a)                                    (b)

**Figure 5.** The wake skewing obtained in the simulations assessed here are compared to results found in literature. Figure 5a shows the actual-to-expected wake skew ratio based on the veer present for the three different wind speeds. Figure 5b shows the skew-to-veer ratio at $8.5D$ downstream and compares it to the fits found for onshore lidar wake measurements performed in Bodini et al. (2017) and LES simulations from Churchfield and Sirnivas (2018). The wake skewing in our simulations is determined by taking the difference between the time-averaged flow wake center at the bottom and top of the rotor plane.

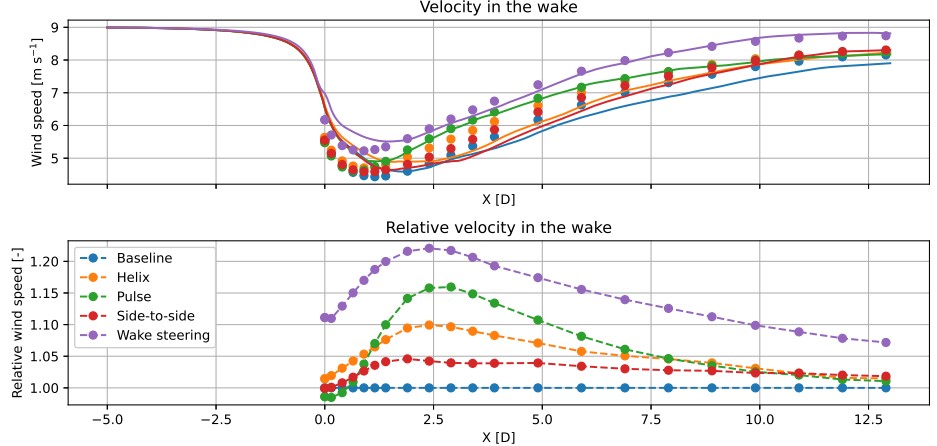

**Figure 6.** The rotor-averaged velocity as a function of downstream distance for the MSLT simulation, assuming alignment with the upstream turbine. The solid lines show the estimated average velocity for all five control strategies when only the streamwise hub height data are taken into account. The dotted line uses cross-sectional planes that capture the entire rotor plane and should therefore be considered more accurate. The relative wind speed in the bottom graph uses the cross-sectional data only.





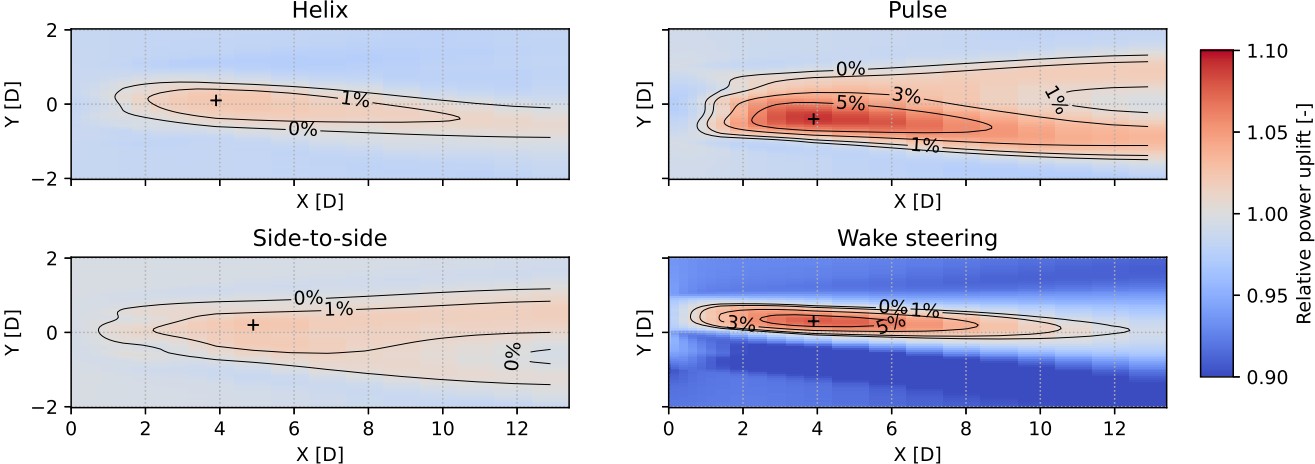

**Figure 7.** The estimated power gains with respect to baseline for a combined two-turbine wind farm when different control strategies are implemented for the MSLT simulation. The power of the downstream turbine is given as a function of its position with respect to the upstream turbine and is based on the rotor-averaged downstream velocities in the cross-sectional planes. The + in each figure shows the location of a downstream turbine that would experience the highest percentage power uplift based on the estimated downstream power.

to the left for wake steering. The helix and side-to-side strategies both have a maximum predicted uplift of 2.4 %, where the side-to-side optimum lies slightly farther downstream.

Note that the width of this uplift region is essentially different for wake steering and wake mixing. For the former, there is a narrow band of turbine locations with high uplift, but outside of this band, wake steering results in substantial power losses. For the latter, the gradient of the uplift with respect to lateral offset is much smaller. This could be an argument for applying

wake mixing in real-world wind farms, as the wind direction is rarely known exactly and varies over time.

Finally, Fig. 7 shows that the IPC-based control strategies (helix and side-to-side) are not nearly as effective as the other two strategies. This is likely related to the skewed shape of the wake: As the wake is not cylindrical, these strategies might not be as successful in creating the helical structures that enable wake mixing in lower-veer conditions. This is further discussed in the next section.

**3.3   Two-turbine results**

In this section, we discuss the results from simulations run with two turbines. The second turbine is located $5D$ downstream of the first turbine with, unless stated otherwise, no lateral offset. First, we discuss the results in the medium-WS, low-TI (MSLT) case, followed by a comparison with the low-veer MSLT-LV and medium turbulence MSMT cases. Finally, we discuss the results from the low- and high-WS cases.





### 3.3.1 Effect of wind farm layout

In this section, we discuss the results from the medium-WS, low-TI (MSLT) case, as defined in Table 3, focusing our attention on the turbine power and DEL of four relevant components through the method described in Sect. 2.4. We study the effect of wind farm layout on the turbine performance by varying the location of the downstream turbine.

For a two-turbine wind farm, with the second turbine aligned with the first $5D$ downstream and running with a baseline controller, the power and critical DELs are plotted in Fig. 8. Looking at the top row, representing the upstream turbine, we see the same power losses as observed in Sect. 3.2 on the $y$-axis, with wake steering resulting in the biggest and side-to-side resulting in the smallest power loss. On the $y$-axes, we see the effect on DEL. What stands out is that wake steering has very minimal effect on the DELs compared to the wake mixing strategies. The blades and tower are most impacted by the pulse strategy while the internal components (yaw bearing and low-speed shaft) are more impacted by the AWM strategies using IPC. The helix and side-to-side strategy generally lead to similar DELs, except for the yaw bearing moments. This can be explained by the lack of rotor tilt moment excitation in the side-to-side method, which subsequently does not excite the yaw bearing pitch moment as the helix strategy does.

The second row shows the results for the downstream turbine. The only strategy that has any substantial impact on the blades is the pulse, although the impact is still significantly lower than the impact on the upstream blades. The downstream tower is most affected by the side-to-side strategy while wake steering reduces the tower DEL of the downstream turbine by a factor of 2. The effect of different control strategies on the internal components is much smaller but again shows that wake steering can lower downstream turbine DELs.

Comparing rows 1 and 2, it should be noted that although some of the turbine 1 DELs increase significantly when AWM is applied, all but the blade loads are still comparable or lower than the loads of the waked second turbine. In other words, although the loads increase, they are still close to the loads of a normally operated waked turbine.

Finally, the third row shows the sum of results from the first two rows. Note that these results are not weighted in any way, and the DELs should therefore not be considered equivalent to the likelihood of damage to any of the two turbines in the wind farm. Clearly, the ideal control strategy would live in the upper-left corner of the plots in this row, as this would represent the highest power capture while the turbines experience the lowest DELs. Wake steering can actually be found closest to this upper-left corner in all the plots, indicating that, in these conditions, wake steering is the superior control strategy.

Next, we study the impact of different wind directions on the effectiveness of the WFC strategies. In these simulations, the downstream turbines are no longer aligned with the wind direction but are instead offset by $-0.5D$ ($120\,\mathrm{m}$) in either direction. This corresponds to a wind direction change of $5.72°$. Note that the yaw angle offset of the upstream turbine is adjusted such that the wake is steered away from the downstream turbine, i.e., for an offset of $0.5D$, the yaw angle is $-20°$ instead of $20°$.

Figure 9 shows the average flow fields when the downstream turbine is offset by $-0.5D$. It demonstrates how, on average, all the wake mixing strategies manage to create a narrower wake, thus increasing the wind speed experienced by the downstream turbine. However, comparatively, the pulse method appears to be the most effective, substantially narrowing the wake from $2.5D$ onward. As expected, though, wake steering is the most effective control strategy when there is a lateral offset in turbine



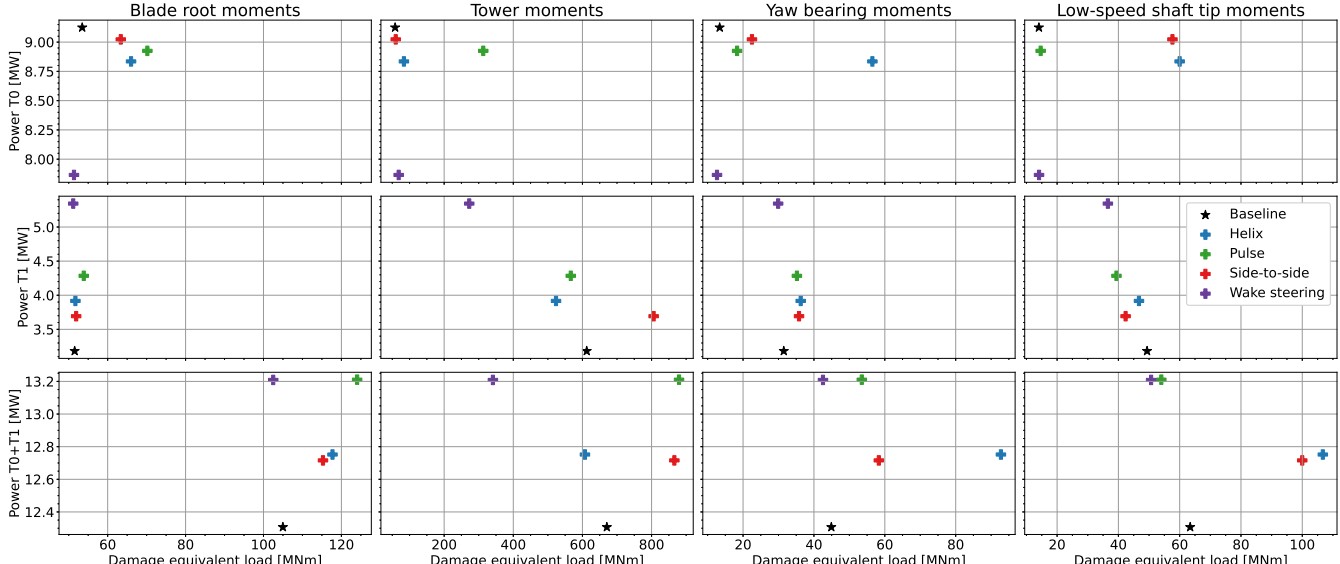

**Figure 8.** DELs plotted on the $x$-axis versus generator power on the $y$-axis for the different control strategies in the medium-WS, low-TI (MSLT) case. The first row shows the results for the upstream turbine, the second row shows the results for the downstream turbine, and the third row sums the results of both turbines. The different columns represent the DEL for different turbine components, as labeled at the top.

locations, as the bottom plot in Fig. 9 shows that the wake is steered almost completely away from the downstream turbine. As
a result, the power uplift is also greatest for wake steering, as shown in Fig. 10.

The power and DELs on turbine 1 are equivalent to those shown in Fig. 8, as the upstream turbine operates in the same conditions regardless of the downstream turbine's location. On the downstream turbine, however, the loads are generally lower than in the aligned case – note that the scales of Figs. 8 and 10 are not identical. As the turbine is now experiencing more less-turbulent freestream flow, the DEL on all components is also reduced substantially. Compared to baseline control, wake
steering now has a slightly adverse effect on the yaw bearing and low-speed shaft DELs, whereas in the aligned case it reduced these loads.

In terms of power production, wake steering is, as expected, clearly the best choice in this scenario, increasing overall power production by 7.8 %. The wake mixing strategies score substantially worse, with 0.9 %, 3.3 %, and 2.0 % power uplift for helix, pulse, and side-to-side strategies, respectively. Note that in these conditions, the side-to-side strategy performs better
than the helix, even though it requires lower pitch actuation and subsequently also has generally lower DELs. This shows that the additional pitch actuation that is intended to excite the wake in the vertical direction is clearly not effective and perhaps even counterproductive in these high-veer, high-shear wind conditions.

Comparing these results to the estimated power gains from Fig. 7, we find that these estimates are close but not exact. The estimated uplift for the helix is very close to the uplift found in the two-turbine simulation, but the side-to-side strategy performs
slightly better than predicted. For the pulse, Fig. 7 predicted a 5 % power uplift, whereas only a 3.3 % uplift is achieved. The





**Figure 9.** Average flow field at hub height for the two-turbine wind farm when different control strategies are applied. The second turbine is offset by -0.5$D$ or 120 m at 5$D$ downstream, equivalent to a wind direction change of 5.72° compared to the aligned case.

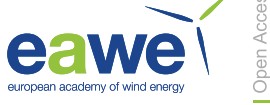
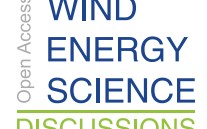


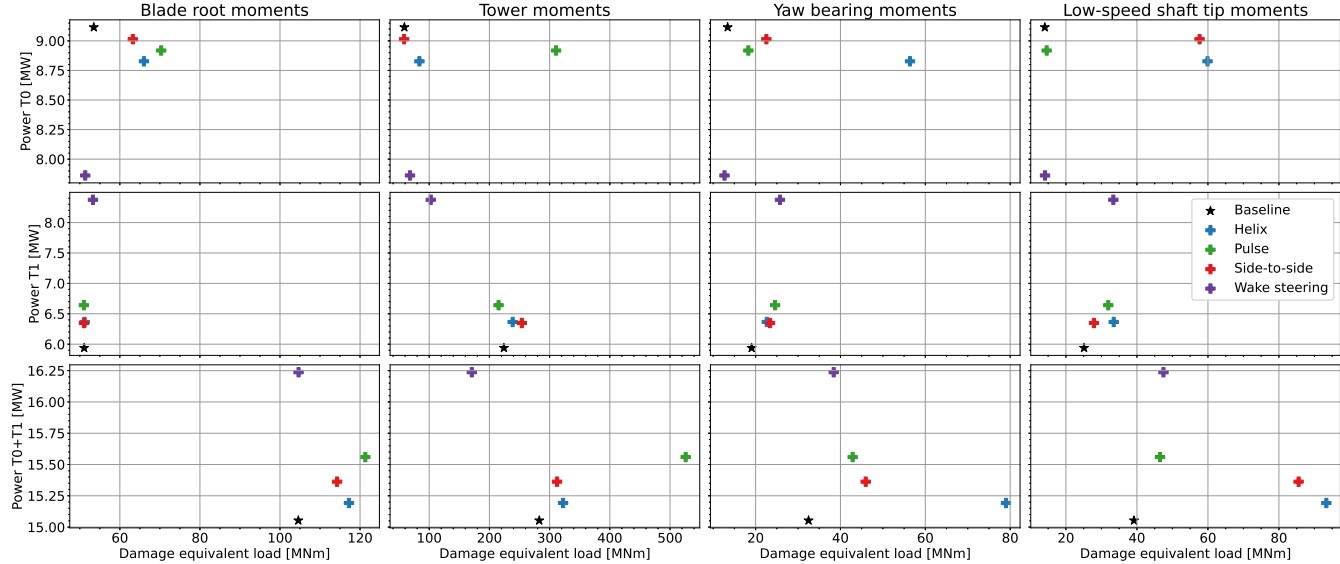

**Figure 10.** DELs plotted on the $x$-axis versus generator power on the $y$-axis for the different control strategies in the medium-WS, low-TI (MSLT) case, with a lateral turbine offset of $-0.5D$. The first row shows the results for the upstream turbine, the second row shows the results for the downstream turbine, and the third row sums the results for both turbines. The different columns represent the DEL for different turbine components, as labeled at the top.

opposite is true for wake steering, which performs substantially better than the 4 % power uplift estimated based on the flow field.

Finally, Fig. 11 shows the power and DELs for a lateral offset of the downstream turbine of $-0.5D$. The first row of this figure is still equivalent to the cases with a different lateral offset, but the second row is remarkably different from the $-0.5D$
offset case. Due to the shape of the wake, shown in Fig. 3, the wake loss at $0.5D$ lateral offset is substantially higher than at $-0.5D$. As a result, wake mixing strategies, specifically the pulse, are able to create more power uplift at the downstream turbine at the same cost for the upstream turbine. This is also visible in Fig. 7, where the negative offset direction clearly shows a higher power uplift than the positive offset direction.

Overall, all strategies still lead to a power uplift compared to the baseline: The helix and side-to-side strategies both increase
power by 1.4 %, the pulse strategy increases power by 10.8 %, and wake steering increases power by 11.0 %. Note that although the power uplift for the IPC-based AWM strategies is limited, they correspond well with the expected uplift predicted in Fig. 7. The pulse strategy, on the other hand, significantly outperforms the expected power uplift obtained from the single-turbine simulations, although they did predict this downstream turbine location to be the "sweet spot" in terms of power uplift.

Investigating the horizontal axes of Fig. 11, we see that wake steering has the lowest DELs for all signals except for the
downstream turbine low-speed shaft. The DELs of the AWM strategies are in line with what we have seen at different lateral offsets. Overall, wake steering should still be considered the best strategy in terms of combined power and loads.





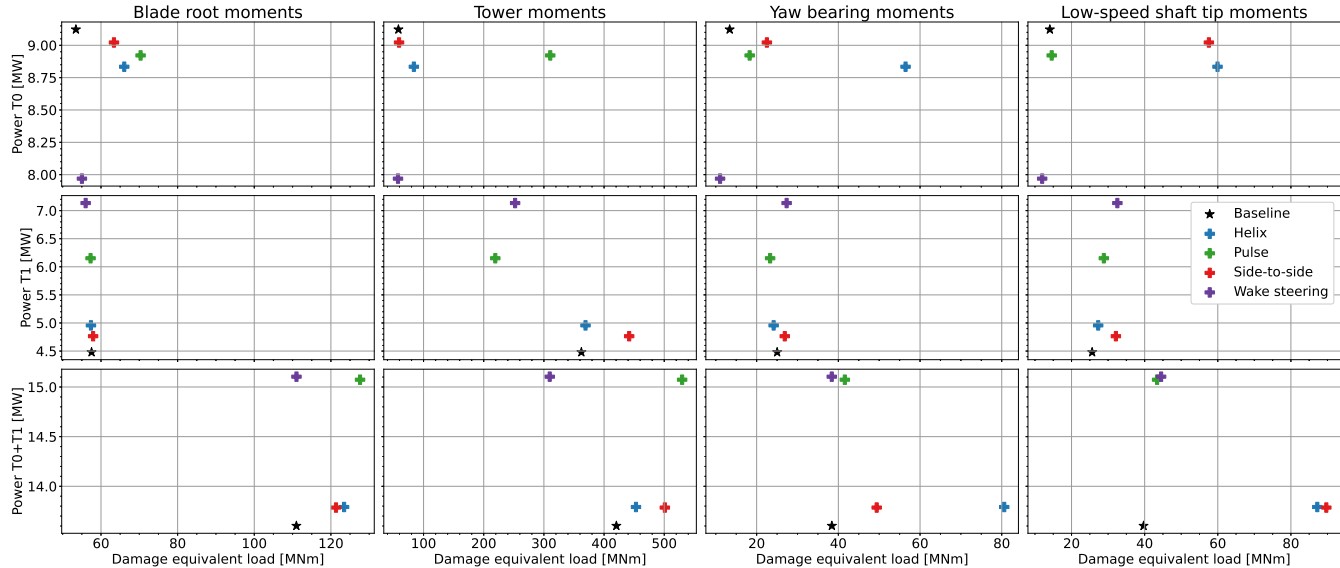

**Figure 11.** DELs plotted on the $x$-axis versus generator power on the $y$-axis for the different control strategies in the medium-WS, low-TI (MSLT) case, with a lateral turbine offset of $0.5D$. The first row shows the results for the upstream turbine, the second row shows the results for the downstream turbine, and the third row sums the results for both turbines. The different columns represent the DEL for different turbine components, as labeled at the top.

### 3.3.2 Effect of veer and turbulence

In this section, we study the results obtained in the cases with low wind veer, as defined in Table 3. First, we consider the results in the MSLT-LV case, which is similar to the conditions discussed in the previous section but with substantially lower wind

veer and shear. Subsequently, the wakes in these conditions are more circular, resulting in higher downstream wake losses. The ensuing power and DEL results are shown in Fig. 12.

If we compare the first row, i.e., the first turbine, with the results in Fig. 8, we observe very similar behavior between the two cases. The overall power is slightly higher in the low-veer case, which makes sense, as the rotor is more uniformly aligned with the wind in this case. The DELs, too, are similar to the high-veer case, with the pulse strategy mostly increasing the tower

DEL, and the helix and side-to-side strategies increasing the DELs of the yaw bearing and low-speed shaft.

The second row of Fig. 12 shows some significant differences with the high-veer case. First and foremost, the power gained at the second turbine when AWM is applied on the upstream turbine is substantially higher here. This likely has to do with the fact that we now have full wake overlap at the downstream turbine, whereas in the previous case, the veered wind already naturally created some recovery at the downstream turbine location (see Fig. 3). Note that the absolute power production of

the downstream turbines for all control cases is still lower than in the MSLT case, but as the baseline power is now lower, the relative power uplift is bigger. Wake steering is still increasing downstream power the most, but the pulse and helix now have a similar uplift. In terms of loads, the differences with the MSLT case are smaller. Most notably, the tower DEL is not decreased



as significantly with wake steering as was the case in Fig. 8. This is likely caused by the fact that the downstream rotor now experiences partial wake overlap, whereas in the MSLT case, the wake was steered away from the downstream turbine more
successfully.

Finally, the bottom row of Fig. 12 displays the sum of the results from the first two rows. Whereas in the MSLT case, this showed wake steering as a clear winner, the picture is a lot more ambiguous in low-veer conditions. Overall, wake steering still performs best in terms of DELs, but is now surpassed in terms of power production by all AWM strategies. The pulse and helix strategies are on par in terms of power production, with, again, the pulse having higher blade and tower loads and the
helix a bigger impact on the yaw bearing and low-speed shaft. In these conditions, picking the most preferable WFC strategy therefore depends on how much priority is given to increasing wind farm power and whether increasing the DEL on certain specific components might be more acceptable than on others.

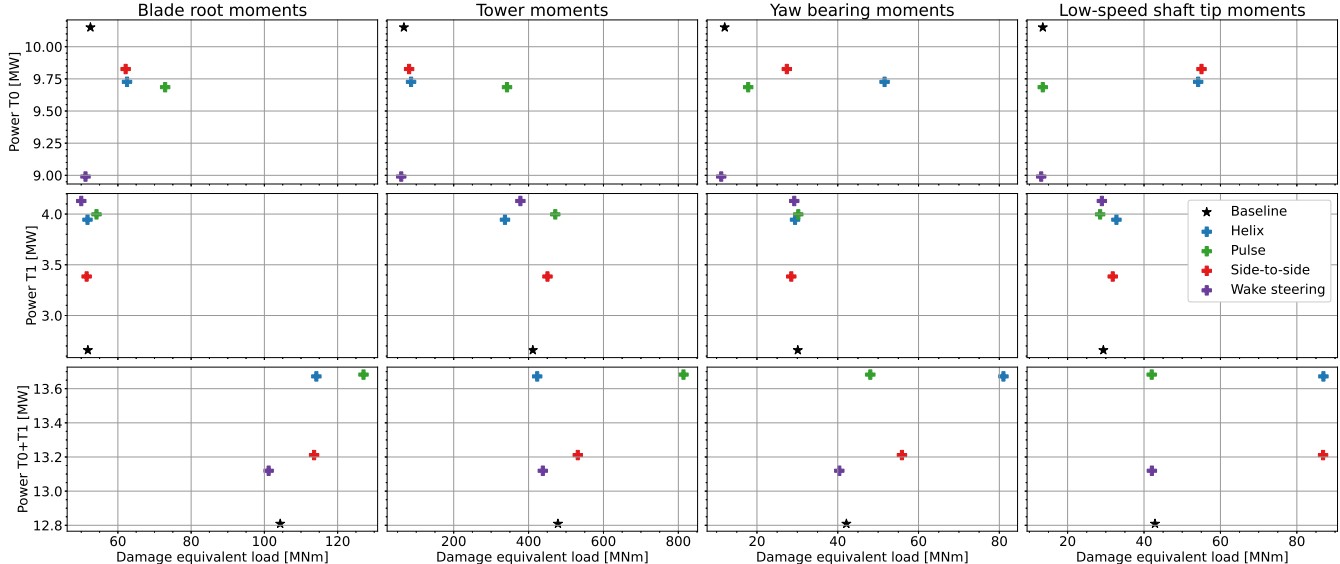

**Figure 12.** DELs plotted on the $x$-axis versus generator power on the $y$-axis for the different control strategies in the medium-WS, low-TI, low-veer (MSLT-LV) case. The first row shows the results for the upstream turbine, the second row shows the results for the downstream turbine, and the third row sums the results for both turbines. The different columns represent the DEL for different turbine components, as labeled at the top.

Next, we study the turbine behavior in the medium-WS, medium-TI (MSMT) case. This case has higher turbulence than any of the other cases studied in this paper, which is expected to have a number of effects on turbine performance. Turbulence
induces natural wake mixing, resulting in lower power deficits at waked downstream turbines. This also affects the potential of WFC, as a lower power deficit means the achievable uplift at the downstream turbine is lower as well. Additionally, variations in wind speed and direction inherent to turbulence are known to result in higher DELs.





The results of these simulations are shown in Fig. 13. The first important observation taken from these plots is that the DELs are, contradictory to literature, lower than in the low-TI case. This is most likely explained by the lower wind veer and shear in the medium-TI simulations. The higher TI causes variations in the moments experienced by the turbine, but at a relatively slow time scale. In the low-TI case, however, the wind veer and shear cause variations at the faster, once-per-rotation ($1P$) frequency, resulting in generally higher DELs. Compared to Fig. 12, the relative change in DELs for all WFC signals is very similar between the low- and medium-TI cases.

A second observation from Fig. 13 is that the relative power uplift achieved by the different WFC strategies matches the low-TI, low-veer case much better than the low-TI, high-veer case. Similar to this case, the pulse and helix strategies are on par as the strategies with the highest uplift, while wake steering results in lower DELs at the expense of lower power production. The relative power uplift compared to the baseline case is, as expected, generally lower than in the low-TI cases. However, for the helix strategy, the power uplift in the MSMT case, 5.2 %, is higher than in the MSLT case (3.6 %). Wake steering and the pulse strategy, on the other hand, only generate half (3.7 %) and two-thirds (5.4 %) of the relative power uplift, respectively, compared to the MSLT case, while the side-to-side method remains about equally effective with a 2.9 % power uplift. This result clearly shows that the effect of veer on WFC effectiveness is substantial, and should be considered when determining the optimal WFC strategy.

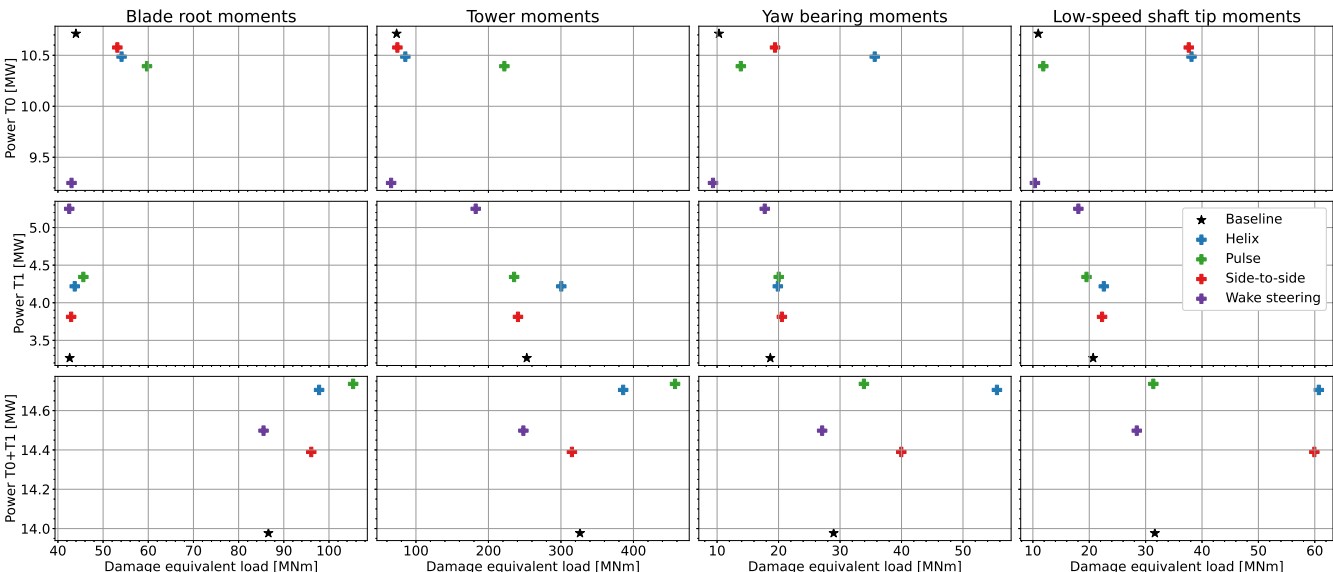

**Figure 13.** DELs plotted on the $x$-axis versus generator power on the $y$-axis for the different control strategies in the medium-WS, medium-TI (MSMT) case. The first row shows the results for the upstream turbine, the second row shows the results for the downstream turbine, and the third row sums the results for both turbines. The different columns represent the DEL for different turbine components, as labeled at the top.





### 3.3.3  Effect of wind speed

Next, we study the results from the low-WS, low-TI (LSLT) case. This case is similar to the MSLT case in a lot of ways. Like
this previously discussed case, it features below-rated wind speed and high wind veer. Subsequently, the results from this case
resemble the MSLT results to a high degree. This is shown in Fig. 14, showing again the power and DELs for the two-turbine
wind farm in these conditions. The most notable difference to the MSLT case is the fact that the pulse strategy loses some of its
ability to increase wind farm power. The DEL results look very similar to the MSLT case, with the most significant difference
being that all control strategies now lead to a general decrease in tower DELs. Subsequently, wake steering is also the preferred
WFC strategy in these simulations, generally reducing overall DELs while resulting in the highest power uplift.

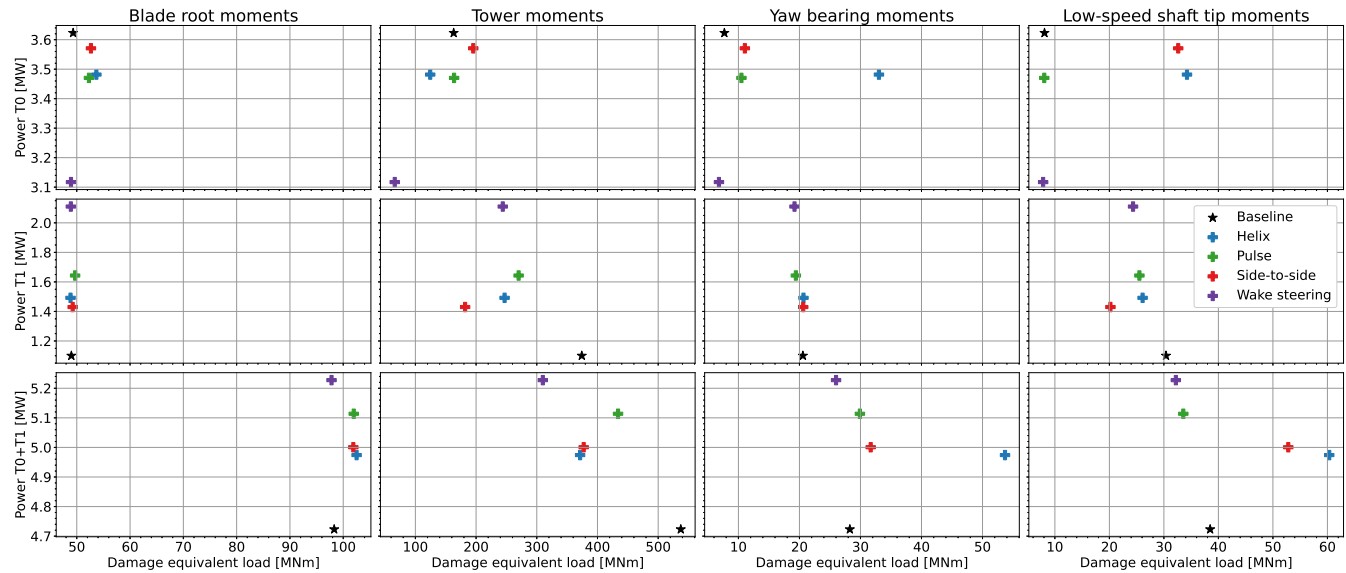

**Figure 14.** DELs plotted on the $x$-axis versus generator power on the $y$-axis for the different control strategies in the low-WS, low-TI (LSLT) case. The first row shows the results for the upstream turbine, the second row shows the results for the downstream turbine, and the third row sums the results for both turbines. The different columns represent the DEL for different turbine components, as labeled at the top.

Finally, we examine the results from the high-WS, low-TI (HSLT) case. As discussed in Sect. 2.3.1, the wind speed is above
rated in this simulation case. This makes this case significantly different from the previous cases. First, applying WFC on the
upstream turbine no longer leads to a loss in turbine power. Therefore, the uplift at the downstream turbine comes without the
cost of power loss at the upstream machine. Second, in the below-rated cases, the baseline pitch angle was constant at zero. In
above-rated conditions, the pitch angle is regulated by the baseline controller to regulate the turbine power to the rated value. The AWM pitch signal is then superimposed on this baseline pitch control signal.

The pulse strategy, in which all blades are pitched collectively, leads to an undesired interaction between the baseline controller and the AWM signal. The fluctuations sent by the active wake controller interfere with the objective of the baseline controller to regulate the power at rated value. As a result, the baseline controller counteracts the AWM signal in order to



keep power constant. This leads to a controller that does not demonstrate the desired variations in thrust that make the pulse a successful WFC strategy. To circumvent this issue, the baseline pitch controller is turned off in the above-rated pulse case. Instead, the baseline controller pitch signal for the pulse simulation is set to be constant at the average angle obtained from the baseline simulation. The AWM signal is now again superimposed on top of this baseline controller signal. Note that this does not lead to behavior that can realistically be expected from an above-rated turbine, and the results from the pulse simulation

should therefore be taken with this in mind. However, for completeness, these results are still included in our analysis.

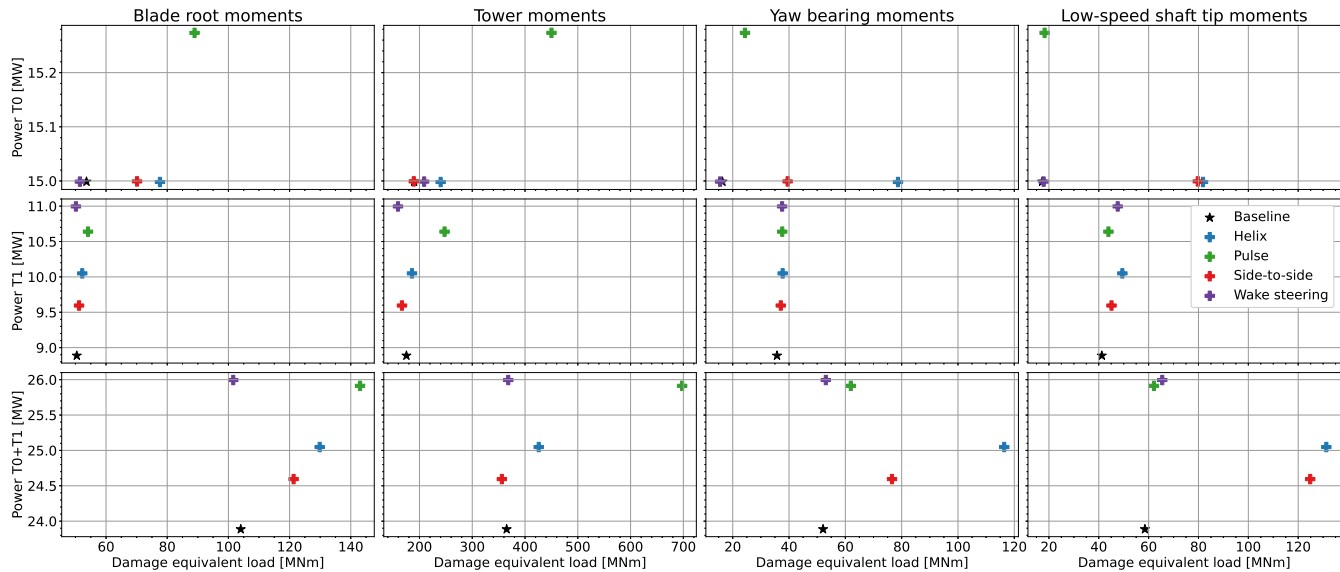

**Figure 15.** DELs plotted on the $x$-axis versus generator power on the $y$-axis for the different control strategies in the high-WS, low-TI (HSLT) case. The first row shows the results for the upstream turbine, the second row shows the results for the downstream turbine, and the third row sums the results for both turbines. The different columns represent the DEL for different turbine components, as labeled at the top. Note that the increased power in the pulse simulation for the first turbine is caused by adjustments made to the controller.

Fig. 15 shows the results for the above-rated conditions. As explained, the power of the first turbine is not truly relevant here. The DELs are affected more significantly in these higher wind speeds, and the baseline DELs are already higher than for any WFC case in the lower wind speed simulations. Overall, the trend is similar to the below-rated simulations, but with wake steering leading to the lowest (if any) increase in DELs. The DELs of the downstream turbine are affected less by the WFC

strategy implemented, with only the tower moments increasing significantly for the pulse case. In these above-rated conditions, wake steering is the most effective strategy in terms of power maximization, surpassing the pulse despite the artificial power uplift the latter sees at the upstream turbine.

The results for all aligned two-turbine simulations investigated in this study are summarized in Table 8 for power and Table 9 for DELs. The power results show that each WFC strategy investigated in this paper is able to increase wind farm power

production in every wind condition studied. Wake steering is generally the most effective strategy in veered wind conditions,





followed closely by the pulse strategy. The helix strategy specifically shows a direct relationship between skewedness of the wake and loss of power uplift.

**Table 8.** The power uplift of each WFC strategy relative to the baseline controller implementation, in each of the wind condition cases studied. Note that the pulse case at HSLT (marked with a *) should be considered academically relevant only, not practically, as the baseline controller was deactivated for the above-rated conditions, resulting in very large power and thrust variations on the upstream turbine.

| Case | Helix | | | Pulse | | | Side-to-side | | | Wake steering | | |
|---|---|---|---|---|---|---|---|---|---|---|---|---|
| Turbine | T1 | T2 | Total | T1 | T2 | Total | T1 | T2 | Total | T1 | T2 | Total |
| LSLT | −3.90 % | +35.64 % | +5.31 % | −4.21 % | +49.36 % | +8.27 % | −1.44 % | +29.96 % | +3.66 % | −13.96 % | +91.81 % | +10.68 % |
| MSLT | −3.18 % | +23.09 % | +3.62 % | −2.18 % | +34.66 % | +7.34 % | −1.11 % | +16.03 % | +3.32 % | −13.08 % | +67.93 % | +7.33 % |
| MSLT-LV | −4.17 % | +48.38 % | +9.47 % | −4.58 % | +50.35 % | +6.82 % | −3.19 % | +27.31 % | +3.14 % | −11.43 % | +55.31 % | +2.42 % |
| MSMT | −2.10 % | +29.20 % | +5.21 % | −2.98 % | +33.02 % | +5.43 % | −1.26 % | +16.79 % | +2.95 % | −13.67 % | +60.80 % | +3.73 % |
| HSLT | +0.00 % | +13.08 % | +4.86 % | +1.83 %* | +19.72 %* | +8.49 %* | +0.00 % | +7.97 % | +2.97 % | +0.00 % | +23.72 % | +8.83 % |

In terms of DELs, the results for all simulations with respect to baseline control is summarized in Table 9. We see here that at lower wind speeds, the impact of using WFC on DELs of the upstream turbine is generally smaller than at higher wind speeds. Furthermore, the impact of a veered wake on the relative DEL change is generally small. Overall, we can conclude that wake steering performs substantially better than the AWM strategies in terms of DELs, with loads generally decreasing or only increasing by a small amount.

## 4 Conclusions and discussion

In this paper (and its companion paper, Brown et al. (2025)), we have investigated and compared the performance of four different WFC strategies that are currently considered in literature as viable options to increase wind farm power. High-fidelity flow simulations were set up in AMR-Wind to replicate average wind speeds, turbulence intensities, and wind veer and shear obtained from lidar measurements in a proposed offshore wind farm site. These datasets exhibited significant wind veer in low-turbulence conditions, which showed to have significant effects on the performance of different WFC strategies.

First, the skewing of the wakes seen in this study were compared to results from literature. This comparison found that the skewing is in line with results from simulations using a different LES code. However, comparing our results with data from onshore lidar measurements shows that these real-world wakes are skewed substantially less relative to the wind veer. It is possible that this is caused by a difference between onshore and offshore conditions, but also that it is a undesired side-effect of how the simulations are set up. More research is therefore needed to conclude which one of these hypotheses holds true.

Regardless of wind speed, the simulations presented in this paper showed that when the wakes are highly skewed due to wake veer, wake steering is generally superior to the different wake mixing strategies investigated. The pulse strategy, where all blades are dynamically pitched collectively, showed similar effectiveness in terms of power uplift, but at the cost of substantially higher DELs. The wake mixing strategies using individual pitch control – the counterclockwise helix and the side-to-side strategy – still achieved a power uplift, but less significant than the former two strategies.





**Table 9.** The relative effect on DEL of each WFC strategy relative to the baseline controller implementation, in each of the wind condition cases studied. All loads are relative to the baseline case for that turbine, where it should be noted that the DELs for the downstream baseline case are generally higher than for the upstream turbine to start with. Note also that the pulse case at HSLT (marked with a *) should be considered academically relevant only, not practically, as the baseline controller was deactivated for the above-rated conditions, resulting in very large power and thrust variations on the upstream turbine.

| DEL | Case | Helix | | Pulse | | Side-to-side | | Wake steering | |
|---|---|---|---|---|---|---|---|---|---|
| | | T1 | T2 | T1 | T2 | T1 | T2 | T1 | T2 |
| **Blade root** | **LSLT** | +8.85 % | −0.32 % | +6.02 % | +1.44 % | +6.78 % | +0.51 % | −0.81 % | −0.20 % |
| | **MSLT** | +23.47 % | +0.36 % | +31.30 % | +4.52 % | +18.53 % | +0.75 % | −3.94 % | −0.73 % |
| | **MSLT-LV** | +19.02 % | −0.21 % | +38.85 % | +4.53 % | +18.30 % | −0.68 % | −2.57 % | −3.49 % |
| | **MSMT** | +22.97 % | +2.61 % | +35.77 % | +7.09 % | +20.84 % | +0.75 % | −2.21 % | −0.25 % |
| | **HSLT** | +44.79 % | +3.55 % | +65.90 %* | +7.27 %* | +30.85 % | +1.45 % | −1.03 % | −0.57 % |
| **Tower base** | **LSLT** | −23.71 % | −34.00 % | +0.42 % | −27.82 % | +19.76 % | −51.38 % | −59.54 % | −34.80 % |
| | **MSLT** | +43.75 % | −14.51 % | +438.19 % | −7.55 % | +3.02 % | +31.59 % | −3.94 % | −55.50 % |
| | **MSLT-LV** | +28.47 % | −18.10 % | +410.77 % | +14.56 % | +20.68 % | +9.50 % | −10.64 % | −8.08 % |
| | **MSMT** | +16.39 % | +18.80 % | +202.60 % | −6.93 % | +1.44 % | −4.82 % | −10.36 % | −27.87 % |
| | **HSLT** | +26.41 % | +6.00 % | +136.95 %* | +41.22 %* | −0.39 % | −4.83 % | +9.97 % | −9.12 % |
| **Yaw bearing** | **LSLT** | +329.10 % | +0.47 % | +36.36 % | −5.61 % | +43.61 % | +0.28 % | −11.54 % | −6.68 % |
| | **MSLT** | +321.45 % | +15.15 % | +36.63 % | 11.74 % | +67.96 % | +13.65 % | −5.62 % | −5.06 % |
| | **MSLT-LV** | +329.30 % | −2.30 % | +48.39 % | +0.36 % | +127.90 % | −5.38 % | −7.20 % | −2.95 % |
| | **MSMT** | +245.98 % | +6.53 % | +34.55 % | +7.31 % | +88.55 % | +10.07 % | −9.37 % | −4.86 % |
| | **HSLT** | +381.03 % | +5.62 % | +48.85 %* | +5.20 %* | +141.46 % | +3.76 % | −5.17 % | +5.07 % |
| **Low-speed shaft** | **LSLT** | +322.83 % | −14.13 % | −0.53 % | −16.05 % | +302.95 % | −33.55 % | −3.06 % | −19.89 % |
| | **MSLT** | +326.59 % | −5.35 % | +4.09 % | −20.23 % | +309.90 % | −14.19 % | +0.31 % | −25.91 % |
| | **MSLT-LV** | +300.77 % | +11.67 % | −0.13 % | −2.88 % | +307.27 % | +8.37 % | −3.15 % | −1.33 % |
| | **MSMT** | +248.26 % | +9.16 % | +8.20 % | −5.67 % | +243.80 % | +7.67 % | −5.40 % | −12.65 % |
| | **HSLT** | +373.20 % | +19.98 % | +5.67 %* | +6.43 %* | +361.03 % | +9.42 % | +3.37 % | +15.45 % |

The study further shows that, regardless of wind conditions, the pulse strategy leads to the most significant increase in DELs
525 on the upstream turbine blades and tower. The yaw bearings and low-speed shaft of the upstream turbine, on the other hand, are more affected by the helix and side-to-side strategies. Comparatively, wake steering has minimal effect on the upstream turbine loads, and in some cases even decreases them. Equivalently, the downstream turbine is affected much less by the control action implemented on the upstream turbine, with generally smaller changes in structural DELs regardless of the control strategy implemented.

530 Additionally, we have executed simulations in otherwise similar wind conditions, but with lower veer and shear. These simulations showed markedly different results in terms of wind farm power uplift, with the pulse and helix wake mixing strategies now being more effective than wake steering. In terms of DELs, the results were similar to the high-veer case.





Simulations at higher turbulence intensity showed similar results. This is more in line with previous studies, which were also executed under low-veer conditions. These results highlight the importance of wind veer, which has so far not been given much attention in WFC simulation research, on the effectiveness of different WFC strategies. The results presented here demonstrate that veer, or more accurately, the wake skewing caused by veer, is a much more important variable to consider than the wind speed or turbulence intensity when choosing the optimal WFC strategy. High-wind-veer conditions generally increase the effectiveness of wake steering while reducing the power uplift achieved with wake mixing strategies that use individual pitch control.

The results presented in this paper raise the question whether wake steering should always be preferred over wake mixing in low-turbulence conditions. The authors stress caution in making such claims. Most of the wakes seen in this study are highly skewed, and it is questionable whether a real-life wake would skew to the same degree. Furthermore, wake steering has another possible downside that is not captured in this study. In real wind farms, the wind direction is constantly changing, and the exact wind direction is not always known instantly or accurately at each turbine. As a result, wake steering sometimes results in a power loss in case of near-perfect alignment, as the wake can be accidentally steered toward a downstream turbine instead of away from it. Furthermore, yaw actuators are not able to respond to changes in wind conditions as quickly as pitch actuators. The effects of time-varying or uncertain wind conditions are not captured in the simulations presented here.

To conclude, this study shows that wind veer plays a major role on how effective different WFC strategies are and should therefore always be considered as a variable when choosing the optimal strategy. Both wake mixing and wake steering strategies can achieve substantial power uplifts in all wind conditions. However, wake steering has shown to be the most reliable tool to achieve wind farm power uplift regardless of wind conditions, at minimal to no cost on the turbine DELs. Nonetheless, to conclusively say that wake steering outperforms wake mixing in realistic wind conditions, additional research is necessary. Future studies should focus on time-varying wind conditions as well as wake skewing analysis and comparison with field measurements. Finally, lidar measurements on the wake of a full-scale turbine operating using different control strategies would be the next step in studying the effects of the control strategies on the wake and therefore on possible downstream turbines.

*Author contributions.* The NREL team (JF, ES, PF) was responsible for the running the majority of the simulations presented in this paper, for executing the analysis, and for writing the paper. Sandia National Laboratories (KB, LC, GY) developed the precursor simulations, and assisted in running simulations and analysis.

*Competing interests.* At least one of the (co-)authors is a member of the editorial board of Wind Energy Science. The authors have no other competing interests to declare.





*Acknowledgements.* This work was authored in part by the National Renewable Energy Laboratory, operated by Alliance for Sustainable Energy, LLC, for the U.S. Department of Energy (DOE) under Contract No. DE-AC36-08GO28308. Funding provided by U.S. Department of Energy Office of Energy Efficiency and Renewable Energy Wind Energy Technologies Office. The views expressed in the article do

565  not necessarily represent the views of the DOE or the U.S. Government. The U.S. Government retains and the publisher, by accepting the article for publication, acknowledges that the U.S. Government retains a nonexclusive, paid-up, irrevocable, worldwide license to publish or reproduce the published form of this work, or allow others to do so, for U.S. Government purposes. A portion of this research was performed using computational resources sponsored by the U.S. Department of Energy's Office of Energy Efficiency and Renewable Energy and located at the National Renewable Energy Laboratory.

570  Sandia National Laboratories is a multimission laboratory managed and operated by National Technology & Engineering Solutions of Sandia, LLC, a wholly owned subsidiary of Honeywell International Inc., for the U.S. Department of Energy's National Nuclear Security Administration under contract DE-NA0003525.



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
