# Peer review of "Comparison of wind-farm control strategies under realistic offshore wind conditions: turbine quantities of interest"

_Wind Energy Science, 2024_

## Author Response (AR1)

**Response to reviewers, Manuscript WES-2024-164**

**Reviewer 1:**

The submitted paper presents a numerical comparison of several Wind Farm Flow Control (WFFC) strategies. Specifically, it investigates the performance of a tandem of wind turbines under different operating scenarios parametrized by inflow velocity and turbulence intensity. The analysis considers power production and Damage Equivalent Loads (DELs) as performance metrics. Alongside the baseline greedy controller, four additional controllers are evaluated: Wake Steering and three Active Wake Mixing (AWM) strategies (Pulse, Helix, and Side-to-Side).

The key finding is that Wake Steering generally outperforms AWM strategies in terms of power production, except under low veer conditions, where AWM yields higher power at the expense of increased loads.

Overall, the paper is well written and provides valuable insights into the impact of WFFC strategies on power production and turbine loads while also highlighting the limitations of the experimentation framework used.

However, several aspects of the paper require further clarification and adjustments, as detailed below. I recommend the publication once these comments have been addressed adequately.

The authors would like to thank the reviewer for their time and efforts in thoroughly reviewing our paper. We appreciate the reviewer's kind words regarding the publication, and hope to have adequately answered the reviewer's concerns in the response below, as well as by the changes made to the manuscript. We believe that the comments provided by the reviewer and our subsequent modifications have made the paper stronger and clearer, and for that we would once again like to express our gratitude towards the reviewer. Please find below our response to each of the individual comments and how we have addressed them in the revised manuscript.

**Major Comments**

1. **Sections 2.2 and 2.3.3:** These sections have some degree of overlap and could be merged inside a single "Turbine control" subsection.

Sections 2.2 and 2.3.3 have now been merged into a new Section 2.3.

2. **Table 4:** The simulations presented in Table 4 have different simulation lengths and cell sizes. Please provide a justification for these choices, especially considering that wake recovery can be sensitive to the mesh resolution.

We presume that the reviewer means Table 1 instead of Table 4, as Table 1 contains the simulation settings. We agree with the reviewer that wake recovery can be sensitive to the mesh resolution, which is why the mesh resolution around the wake is the same (2.5 m) for all simulation cases. We acknowledge that this was not sufficiently emphasized in the original manuscript, so we have added a paragraph detailing this. Some of the other differences in settings are practical, chosen to assure the desired wind conditions within the simulation window while managing computation time. More details on how these precursor simulations were set up are provided in Section 2.3 of the companion paper (Brown, et al., 2025). A preprint of this paper can be found here.

3. Line 212: Given that loads are considered in the study, more details on the specific OpenFAST settings used are required.

We agree that this was lacking in the original version of the manuscript. Two paragraphs have been added to the manuscript describing the settings and modules used for the OpenFAST simulations.

4. **Table 4:** This table could be removed. It would recommend to reference the original publication and, instead, plot the power and thrust curves, along with the three wind speeds considered in this study.

Table 4 has been removed from the manuscript. However, following the reviewer's logic, the power and thrust curves can also be found in the reference, and are therefore not added to the manuscript. Some highly relevant parameters, such as hub height, rotor diameter, and rated wind speed, have been added to this section.

5. Line 236: The choice of yaw angle must be justified. The performance of yaw steering strategies (in terms of both power and loads) varies significantly with the yaw offset chosen. Clarify how the yaw angle was selected and demonstrate that it achieves the desired wake offset.

We fully agree with the reviewer that the performance of wake steering through yaw varies significantly with yaw offset chosen. On the other hand, the same can be said about AWM. For both strategies it holds that larger offsets from baseline result in higher power gains at the expense of higher loads. Therefore, there's no such thing as an absolute "optimum" in terms of yaw offset. However, we agree that the choice for a 20-degree offset is not justified in the original manuscript. We therefore added the following paragraph to the paper around line 236:

"Equivalently to the AWM pitch amplitude, in the simplified case of a constant wind direction with full wake overlap, a larger yaw angle offset generally results in higher powers at the expense of higher loads. Therefore, to match the large pitch amplitudes used in AWM, we have also chosen to implement a relatively large yaw angle offset. In field experiments, a yaw angle offset of up to 20° is commonly considered acceptable to manage structural loads (Fleming, et al., 2019; Doekemeijer, et al., 2021; Simley, et al., 2021). We have subsequently chosen to implement a yaw angle offset of ±20° in all WS simulations."

6. **Line 247:** The phrase "a constant yaw alignment with respect to the mean wind direction" is ambiguous. In practice, defining the freestream velocity is not straightforward. Clarify whether the wind direction is assumed to be known or if it is dynamically measured by the simulated turbines.

We agree that the original phrasing was ambiguous. We have added the following lines to resolve this ambiguity: "Note that, for simplicity, we assume that the wind direction is constant over time and known a priori in all control cases. Subsequently, the yaw angle is constant over the entire simulation and based on the average wind direction."

7. **Precursor Setup:** Given the critical role of veer in the study, the precursor simulation setup must be described in detail. Include the velocity profile generated and any adjustments made to match inflow conditions.

The precursor simulation setup is described in detail in Section 2.3 of the companion paper (Brown, et al., 2025), and this paper is also referenced in the section of this paper discussing the precursor setup. We have chosen not to copy this information directly into this paper too, as we feel like we make it very clear on multiple occasions that these two papers are companions, and both papers are already quite long without repeating much of the same information. However, we realize that, due to delays in the submission of the companion paper, this paper was not publicly available yet at the time of the review. We hope that the reviewer, upon reading the companion paper now that it is available, agrees that this paper describes the setup in sufficient detail.

Figure 7: As expected, the power uplift for AWM strategies climbs back to ~100% away from the wake. However, it is unclear why this is not the case for wake steering. Please review this figure or provide an explanation for this behavior.

Figure 7 is accurate. As the caption and the accompanying text clearly indicates, it shows the estimated power gains for a *combined* two-turbine wind farm. It therefore considers the power losses compared to baseline at the upstream turbine when the different control strategies are implemented. With AWM, this power loss is minimal,

resulting in a minimal total power loss when there is no wake interaction between the upstream and downstream machine. However, for wake steering, this power loss is much more significant (see Figure 2). As a result, in the case of no wake interaction, the power losses for wake steering are substantial, and the two-turbine power uplift is negative.

Although we do discuss this difference in Section 3.2.2, we now realize that this might not have been clear to all readers. We therefore added a couple of lines to this section outlining what we explained above.

9. Figures 8, 10, 11, 12, 13, 14, and 15: The baseline controller's performance should be more visible in these plots. Consider adding vertical and horizontal lines to highlight its location. Additionally, the font size for axis labels is too small. To facilitate comparison, I recommend merging some of these figures (e.g., cases 0.5D and - 0.5D).

Horizontal and vertical lines have been added to show the location of the baseline case with respect to the different control strategies. The font sizes have been increased, and the two offset cases have been merged into a single figure.

**Minor Comments**

1. Line 30, Line 43, and subsequent occurrences: The term "set point" should be written as "setpoint".

We feel like this is one of those cases where the English language lacks a "right" and "wrong", as a quick Google search shows that the internet does not seem to agree on an absolute truth. Nevertheless, we agree that "setpoint" is less prone to confusion than "set point" and have therefore changes this throughout the manuscript.

2. Line 31: The term "Wind Farm Control" is too generic. Consider using "Wind Farm Flow Control" (WFFC) as defined by Meyers et al. (2022). "We define WFFC as the coordinated control of the turbines in the farm, with the aim of influencing the flow (wakes, turbulence) in such a way that it improves the overall figure of merit of the farm"

We agree and have change "WFC" to "WFFC" throughout the manuscript.

3. Line 53: The phrase "allow or even depend" is confusing.

This phrase has been altered to "not only allow, but even depend on".

4. Line 145: To differentiate vector notation from script, consider using the \mathbf notation.

This suggestion has been implemented.

5. **Table 1:** The column heading "Desired Effect" is not ideal, as the desired effect is always to increase global power production. Consider a more descriptive heading.

We would argue that this is a *very* high-level effect, and that we are quite obviously referring to a lower-level effect. To point this out more explicitly, we have changed the header to "Desired effect *on the wake*".

6. Line 231 and subsequent: A = 4 deg, degree need to be specified.

A degree symbol has been added.

7. Line 233: A reference is missing for the statement on the selection of yaw offset angles.

This reference has been fixed.

8. **Figures 3 and 4:** Consider merging these figures to facilitate comparison. Ensure the font size is consistent and readable across all figures.

The font size is consistent and readable across all figures. We feel that merging these figures is not desirable, but have placed them on the same page to facilitate comparison.

9. **Figure 5:** Add a vertical line at 8.5D to indicate the location from which the data is extracted.

A vertical line has been added at 8.5D in Figure 5a.

10. **Line 385:** The statement about narrower wakes resulting in higher power production for the downstream turbine is valid. However, the narrowest wake would be achieved by shutting down the upstream turbine entirely. Please clarify that the aim is to achieve a narrower wake while maintaining a similar power output for the upstream turbine.

A clause has been added here specifying that the narrowing wake is achieved while losing minimal power at the upstream turbine.

**Reviewer 2:**

**General Comments**

The paper presents interesting and highly relevant research on the effectiveness of different wind farm control methods, with special attention paid to the effect of veer.

Overall, the paper is of high scientific quality and the methodology is sound, clear and explicit.

The results are presented in a clear manner and the conclusions drawn are well supported by the presented results.

The authors sincerely thank the reviewer for their kind words regarding this paper, as well as for taking the time to review it. We strongly feel that the changes we have made based on the reviewer's comments have helped us further improve and finetune the manuscript and are confident that we have addressed all the concerns raised by the reviewer in doing so. Please find below our response to each of the individual comments and how we have addressed them in the revised manuscript.

**Major Comments**

I believe the studied LES simulation requires a more detailed description. The equations used and common quantities defining the inflow, such as roughness length, inversion height, coriolis parameter, etc. are missing. Also, the link provided in the reference list doesn't work. Similarly, the used turbine model should be described in more detail, such as what kind of solver is used (elastodyn, beamdyn) in openFAST and which degrees of freedom are activated.

The authors agree that this information is required for the reader to be able to understand and reproduce our analysis. In terms of the OpenFAST settings, we have added information on the blade solver and degrees of freedom that were used during the simulations. Due to the quantity and computational load of the simulations performed for this paper, we have selected to use ElastoDyn instead of BeamDyn. We are fully aware that this reduces the fidelity of the blade load signals, and have also added a paragraph describing these limitations in Section 2.2, which describes the turbine model.

Regarding the LES simulations: these descriptions were deliberately kept short. The creation of these precursors is part of the work presented in this paper's companion (Brown, et al., 2025). We felt like it was unnecessary and undesirable to repeat the exact same information already presented in the companion paper, and therefore felt like referring to it sufficed. We realize however that at the time of this review, this paper was not publicly available yet due to some delays in its submission process. Now that it can be found here, we trust that the

reviewer agrees that the description given there is adequately elaborate and it therefore sufficient to refer to it in this paper.

Furthermore, I think the methodology of chapter 3.2.2 can be improved by considering a different method to calculate the power of a virtual turbine, for example by using the rotor equivalent wind speed (Wagner, R., Courtney, M., Gottschall, J., Lindelöw-Marsden, P., 2011. Accounting for the speed shear in wind turbine power performance measurement. Wind Energy 14, 993–1004), instead of current equation 3. Furthermore, I believe there are some typos in equations 3 and 4.

The typos have been corrected, and we have investigated the alternative methodology proposed by the reviewer. The method of rotor equivalent wind speed by Wagner et al is mainly designed to account for lidar measurements at different heights, and the difference in rotor swept areas at these heights. In our simulations, we have access to wind speed data over a relatively fine, evenly distributed grid around the rotor swept area. Using the average of all these grid points is effectively equivalent to averaging at different heights and subsequently multiplying by the rotor swept area at each height. Nonetheless, we have implemented this methodology and used it to regenerate Figure 7. The result is shown below. When this Figure is compared to the original found in the manuscript, the differences are shown to be small to the point of being impossible to perceive. We believe this confirms our original results, and therefore justifies sticking to the original methodology described in the manuscript.

Figure 1: A re-rendering of Figure 7 of the original manuscript, using the rotor equivalent wind speed method suggested by the reviewer.

I also find the notation used in section 2.1 not entirely clear. I suggest to clarify the difference between indices and name-particles by using italicized and non-italicized subscripts, respectively. (I especially stumbled over \$u\_{h,j,k}\$, is \$h\$ supposed to be the index of the point in x at hub height?)

The reviewer is correct in their presumption that the *h* in these equations stands for the index of the *x*-location of the virtual turbine rotor hub. We have tackled this lack of clarity by now specifically defining this subscript, as well as making it non-italicized as suggested by the reviewer.

I find there are some limitations in the conclusions that could be addressed by the authors. Specifically, using a wind farm of two turbines and applying the control mechanism only at the first turbine limits the applicability of the findings. I believe this is especially relevant in the case of wake steering, as here the power losses at the actuated turbine are significant. Presumably, wind farm control would be applied to the first few rows in a large wind farm (see, for example Howland, M.F., Lele, S.K., Dabiri, J.O., 2019. Wind farm power optimization through wake steering. PNAS 116, 14495–14500). Hence, reductions in power would also be observed at further downstream turbines. The authors should, at least, discuss this limitation, or, even better, provide additional results to quantify the difference.

We fully agree with the reviewer that the limitations of our simplified setup have not been addressed sufficiently in our original manuscript. We have therefore added a paragraph to Section 2.1, discussing why we chose this simple setup and what the limitation of this choice are. Indeed, in a larger wind farm, downstream turbines could possibly be controlled as well. Note that, although more established in the case of wake steering, this applies to wake mixing strategies too (see, e.g., van Vondelen, Ottenheym, Pamososuryo, Navalkar, & van Wingerden, 2023).

Our current setup already requires a minimum of 5 different simulations for each wind case, each of which take in the order of 40.000 core-hours to run on our high performance computing systems. Including potential control actions on downstream turbines would at minimum double the number of simulations, in addition to increasing their individual computational load. Furthermore, finding appropriate control signals for downstream turbines at different control settings, or the validation of a lower-fidelity model to our simulation data. We do not have the required computational capacity for the former, while the latter is beyond the scope of our paper.

As the reviewer noted in their general comments, the novelty of the simulations presented in these research lies primarily in studying the effects of different wind farm control strategies in varying wind conditions, and specifically on the effect of wind veer. As this is a completely novel direction that, as far as the authors are aware, has been ignored in similar studies so far, we believe that it is justified—and perhaps even desirable—to focus on a simple wind farm layout case. That does not take away the fact that studying effects deeper inside the wind farm are very relevant. We have therefore added this to our conclusions as a possible direction for future work.

Furthermore, the authors discuss the importance of uncertainty in wind direction on wake steering. I suggest to refer to work by Taschner et al. (Taschner, E., Becker, M., Verzijlbergh,

R., Van Wingerden, J., 2024. Comparison of helix and wake steering control for varying turbine spacing and wind direction. J. Phys.: Conf. Ser. 2767, 032023).

The work mentioned was originally already referenced in the introduction, but we agree that due to the similarities to some of the analyses executed in our paper, it justifies an additional reference in the Results section. We have added a reference in Section 3.2.2, including a short summary of the findings in this paper regarding wind direction changes.

In section 3.3.1 the authors write that the AWM achieve a narrowing of the wake. I would argue that this narrowing is a combined effect of the time-averaging and enhanced mixing. As this is not the focus of the paper I don't suggest to go into detail but rather rephrase in a more precise manner.

We fully agree that this is also a result of the time-averaging. However, as both the sentence where we argue the narrowing wake and the preceding sentence mention the averaging that is carried out here, we feel like this is sufficiently concise and rephrasing is therefore unnecessary. We have however added the note that the averaging is done over time.

Finally, in the abstract and introduction the authors refer to the companion paper, however, in the rest of the paper no references are made. It would be nice if the authors linked their observations to the companion paper throughout the article. For example, in line 370 the author point out that the tower loads are most significantly affected by the side-to-side strategy. It would be nice to offer an explanation, or link to the companion paper if an explanation can be found there.

Although there are numerous references to the companion paper after the introduction, we admit that we could have made more references to link the results from this paper to those seen in the companion paper. We have done so in the updated manuscript, including in line 370. Note that we now refer to the companion paper a total of 9 times throughout the paper.

**Minor comments**

l. 1: ... and therefore lower the levelized cost of energy, [of] a wind ...

**This has been corrected.**

l. 163: In my opinion, \$C^\mathrm{opt}\_\mathrm{P}\$ should not depend on \$\lambda\$

**The dependency has been removed.**

l. 231: missing ° for the amplitude

**Symbol has been added.**

l. 232: missing reference (rendered only as a ?)

This reference has been fixed.

Figure 1: I believe labels for side-to-side and pulse are mixed up

It is interesting how, somehow, 10 experts (all 6 authors, 3 internal reviewers, and the first external reviewer) all thoroughly review a paper, but not one of them spots a mix up as obvious as this one; you are obviously right. We thank you for your diligence and have corrected the error.

l. 258: \$m\$ is described as the slope of the Wöhler curve, but in table 6 its called the Wöhler exponent. I suggest to use the same description in both instances.

The parameter *m* is now uniformly called the Wöhler exponent.

Figure 3: the x-axis label is X, but I believe it should be Y and the y-axis label is Y but should be z, similarly in the caption the hub height is given as y=... but it should be z.

**This error has been fixed.**

Figure 4: same as figure 3, also the upper plot is referred to as a velocity profile but I would call it a velocity field or slice and not a profile.

We choose not to change this, as we think the current title is clear. We have fixed the label errors.

Figure 5 (b): the slope marking 1:1 veer is green, but was black in 5a

This was done to have the figures correspond to the figure shown in the respective references, but we can see how that might be confusing. We have therefore changed the slope to be black in both figures.

Figure 6: in the upper figure, only round markers mark the rotor average velocity planes, while a dashed line with round markers is used in the lower plot. I suggest to use the same style in both plots to emphasize to the reader that they are based on the same calculation.

We believe the caption makes it sufficiently clear what both figures and lines show. We have chosen not to include a dashed line in the upper figure, or remove the dashed line in the lower figure, as this reduces the readability of the figure overall.

Figure 9: this figure is very large. Maybe the figure size can be reduced by showing less of the region upstream of the first turbine.

**We have altered this figure to take up less space.**

Table 8: Maybe a visual representation of this overview, such as a heatmap, is easier to digest. Similarly table 9.

We agree with the reviewer that these tables are not very visually pleasing. However, we do not see the need to make these tables into a visual representation, as these tables are already visually presented in Figure 8 and Figures 10-14. These tables are meant serve as a reference for readers who are interested in the exact value of a specific power or load channel in specific wind conditions. We therefore choose to keep the tables as they are.

**References**

- Brown, K., Yalla, G., Cheung, L., Frederik, J. A., deVelder, N., Houck, D., . . . Fleming, P. (2025). Comparison of wind-farm control strategies under realistic offshore wind conditions: wake quantities of interest. *Wind Energy Science*.
- Doekemeijer, B. M., Kern, S., Maturu, S., Kanev, S., Salbert, B., Schreiber, J., . . . Wilts, F. (2021). Field experiment for open-loop yaw-based wake steering at a commercial onshore wind farm in Italy. *Wind Energy Science*.
- Fleming, P., King, J., Dykes, K., Simley, E., Roadman, J., Scholbrock, A., . . . Fleming, K. (2019). Initial results from a field campaign of wake steering applied at a commercial wind farm--Part 1. *Wind Energy Science*.
- Simley, E., Fleming, P., Girard, N., Alloin, L., Godefroy, E., & Duc, T. (2021). Results from a wake-steering experiment at a commercial wind plant: investigating the wind speed dependence of wake-steering performance. *Wind Energy Science*.
- van Vondelen, A., Ottenheym, J., Pamososuryo, A. K., Navalkar, S. T., & van Wingerden, J.-W. (2023). Phase synchronization for helix enhanced wake mixing in downstream wind turbines. *IFAC*.

---

## Referee Report (RR1)

**Review**

**General Comments**

My comments have been addressed in a satisfactory manor and I believe the paper is now ready for publication.

However, I have found some minor editorial changes:

l.118: the effect they have [on] the turbine

Table 3: column 3, row 3, it says the pulse modulates the wake depth, I think it should say wake width

l. 251: missing unit of the amplitude

Figure 3, caption: it should be z/D=0.625

Figure 4: same as figure 3.